# Achieving Forgetting Prevention and Knowledge Transfer in Continual Learning

**Zixuan Ke**[1]**, Bing Liu**[1]**, Nianzu Ma**[1]**, Hu Xu**[2] **and Lei Shu**[3*]
[1]Department of Computer Science, University of Illinois at Chicago
[2]Facebook AI Research
[3]Amazon AWS AI
[1]`{zke4,liub,nma4}@uic.edu`
[2]`huxu@fb.com`
[3]`shulindt@gmail.com`

## Abstract

Continual learning (CL) learns a sequence of tasks incrementally with the goal of achieving two main objectives: *overcoming catastrophic forgetting* (CF) and *encouraging knowledge transfer* (KT) *across tasks*. However, most existing techniques focus only on overcoming CF and have no mechanism to encourage KT, and thus do not do well in KT. Although several papers have tried to deal with both CF and KT, our experiments show that they suffer from serious CF when the tasks do not have much shared knowledge. Another observation is that most current CL methods do not use pre-trained models, but it has been shown that such models can significantly improve the end task performance. For example, in natural language processing, fine-tuning a BERT-like pre-trained language model is one of the most effective approaches. However, for CL, this approach suffers from serious CF. An interesting question is how to make the best use of pre-trained models for CL. This paper proposes a novel model called CTR to solve these problems. Our experimental results demonstrate the effectiveness of CTR.[2]

## 1 Introduction

This paper studies continual learning (CL) of a sequence of natural language processing (NLP) tasks in the *task continual learning* (Task-CL) setting. It aims to (i) prevent catastrophic forgetting (CF), and (ii) transfer knowledge across tasks. (ii) is particularly important because many tasks in NLP share similar knowledge that can be leveraged to achieve better accuracy. CF means that in learning a new task, the existing network parameters learned for the previous tasks may be modified, which degrades the performance of previous tasks [40]. In the Task-CL setting, the task id is provided for each test case in testing so that the specific model for the task in the network can be applied to classify the test case. Another popular CL setting is *class continual learning*, which does not provide the task id during testing but it is for solving a different type of problems.

Most existing CL papers focus on dealing with CF [21, 5]. There are also some papers that perform knowledge transfer. To achieve both objectives is highly challenging. To overcome CF in the Task-CL setting, we don't want the training of the new task to update the model parameters learned for previous tasks to achieve model separation. But to transfer knowledge across tasks, we want the new task to leverage the knowledge learned from previous tasks for learning a better model (forward transfer) and also want the new task to enhance the performance of similar previous tasks (backward transfer).

---

[*]Work was done prior to joining Amazon.
[2]The code of CTR can be found at `https://github.com/ZixuanKe/PyContinual`

35th Conference on Neural Information Processing Systems (NeurIPS 2021).

This means it is necessary to update previous model parameters. This is a dilemma. Although several papers have tried to deal with both [22, 37], they were only tested using sentiment analysis tasks with strong shared knowledge. When tested with tasks that don't have much shared knowledge, they suffer from severe CF (see Sec. 5.4). Those existing papers that focus on dealing with CF do not do well with knowledge transfer as they have no explicit mechanism to facilitate the transfer.

Another observation about the current CL research is that most techniques do not use pre-trained models. But such pre-trained models or feature extractors can significantly improve the CL performance [18, 24]. An important question is how to make the best use of pre-trained models in CL. This paper studies the problem as well using NLP tasks, but we believe that the developed ideas are also applicable to computer vision tasks because most pre-trained models are based on the transformer architecture [60]. We will see that the naive or the conventional way of directly adding the CL module on top of a pre-trained model is not the best choice (see Sec. 5.4).

In NLP, fine-tuning a BERT [8] like pre-trained language model has been regarded as one of the most effective techniques in applications [65, 57]. However, fine-tuning works poorly for continual learning. This is because the fine-tuned BERT for a task captures highly task-specific information [41], which is difficult to be used by other tasks. When fine-tuning for a new task, it has to update the already fine-tuned parameters for previous tasks, which causes serious CF (see Sec. 5.4).

This paper proposes a novel neural architecture to achieve both CF prevention and knowledge transfer, which also deals with the CF problem with BERT fine-tuning. The proposed system is called **CTR** (*Capsules and Transfer Routing for continual learning*). CTR inserts a continual learning plug-in (CL-plugin) module in two locations in BERT. With the pair of CL-plugin modules added to BERT, we no longer need to fine-tune BERT for each task, which causes CF in BERT, and yet we can achieve the power of BERT fine-tuning. CTR has some similarity to Adapter-BERT [16], which adds adapters in BERT for parameter efficient transfer learning such that different end tasks can have their separate adapters (which are very small in size) to adapt BERT for individual end tasks and to transfer the knowledge from BERT to the end tasks. Then, there is no need to employ a separate BERT and fine-tuning it for each task, which is extremely parameter inefficient if many tasks need to be learned. An adapter is a simple 2-layer fully-connected network for adapting BERT to a specific end task. A CL-plugin is very different from an adapter. We do not use a pair of CL-plugin modules to adapt BERT for each task. Instead, CTR learns all tasks using only one pair of CL-plugin modules inserted into BERT. A CL-plugin is a full CL network that can leverage a pre-trained model and deal with both CF and knowledge transfer. Specifically, it uses a *capsule* [15] to represent each task and a proposed *transfer routing* algorithm to identify and transfer knowledge across tasks to achieve improved accuracy. It further learns and uses task masks to protect task-specific knowledge to avoid forgetting. Empirical evaluations show that CTR outperforms strong baselines. Ablation experiments have also been conducted to study where to insert the CL-plugin module in BERT in order to achieve the best performance (see Sec. 5.4).

## 2   Related Work

**Catastrophic Forgetting:** Existing work in CL mainly focused on overcoming CF using the following approaches. (1) *Regularization-based approaches,* such as those in [27, 30, 51, 69], add a regularization in the loss to consolidate weights for previous tasks when learning a new task. (2) *Replay-based approaches*, such as those in [45, 36, 4, 63], retain some training data of old tasks and use them in learning a new task. The methods in [54, 20, 47, 14] learn data generators and generate old task data for learning a new task. (3) *Parameter isolation-based approaches,* such as those in [52, 21, 39, 10], allocate model parameters dedicated to different tasks and mask them out when learning a new task. (4) *Gradient projection-based approaches* [68] ensure the gradient updates occur only in the orthogonal direction to the input of old tasks and thus will not affect old tasks. Some recent papers used pre-trained models [18, 23, 24] and learn one class per task [18]. Tackling CF only deals with model deterioration. These methods perform worse than learning each task separately. An empirical study of the cause of CF and the impact of task similarity on CF was done in [44].

Some NLP applications have also dealt with CF. For example, CL models have been proposed for sentiment analysis [23, 24, 37, 43], dialogue slot filling [53], language modeling [58, 7], language learning [31], sentence embedding [33], machine translation [25], cross-lingual modeling [35], and question answering [12]. A dialogue CL dataset is also reported in [38].

**Knowledge Transfer:** Ideally, learning from a sequence of tasks should *also* allow multiple tasks to support each other via knowledge transfer. CAT [21] (a Task-CL system) works on a mixed sequence of similar and dissimilar tasks and can transfer knowledge among similar tasks detected automatically. Progressive Network [48] does forward transfer but it is for class continual learning (Class-CL).

Knowledge transfer in this paper is closely related to *lifelong learning* (LL), which aims to improve the new/last task learning without handling CF [56, 49, 5]. In the NLP area, NELL [3] performs LL information extraction, and several other papers worked on lifelong document sentiment classification (DSC) and aspect sentiment classification (ASC). [6] and [61] proposed two Naive Bayesian methods to help improve the new task learning. [64] proposed a LL approach based on voting. [55] used LL for aspect extraction. [43] and [62] used neural networks for DSC and ASC, respectively. Several papers also studied lifelong topic modeling [5, 13]. However, all these works do not deal with CF.

SRK [37] and KAN [22] try to deal with both CF and knowledge transfer in continual sentiment classification. However, they have two critical weaknesses: (i) Their RNN architectures cannot use plug-in or adapter modules to tune BERT, which significantly limits their power. (ii) Since they were mainly designed for knowledge transfer, they suffer from serious CF (see Sec. 5.4). B-CL [24] uses the adapter idea [16] to adapt BERT for sentiment analysis tasks, which are similar to each other. However, since its mechanism of *dynamic routing* for knowledge transfer is very week, its knowledge transfer ability is markedly poorer than CTR (see Sec. 5.4). CLASSIC [23] is another recent work on continual learning for knowledge transfer, but its CL setting is *domain continual learning*. Its knowledge transfer method is based on contrastive learning.

AdapterFusion [42] used adapters proposed in [16]. It proposes a two-stage method to learn a set of tasks. In the first stage, it learns one adapter for each task independently using the task's training data. In the second stage, it uses the training data again to learn a good composition of the learned adapters in the first stage to produce the final model for all tasks. AdapterFusion basically tries to improve multi-task learning. It is not for continual learning and thus has no CF. As explained in Sec. 1, the CL-plugin concept in CTR is different from that of adapters for adapting BERT for each task. CL-plugins are continual learning systems that make use of a pre-trained model.

## 3 CTR Architecture

This section describes the general architecture of CTR. The details about its key component CL-plugin is presented in the next section. Due to its good performance, BERT [8] and its transformer [60] architecture are used as the base in our model CTR. Since BERT fine-tuning is prone to CF (Sec. 1), we propose the CL plug-in idea, which is inspired by Adapter-BERT [16]. CL-plugin is a full continual learning module designed to interact with a pre-trained model, in our case, BERT.

**Inserting CL-plugins in BERT.** A commonly used method of leveraging a pre-trained model is to add the end task module on top of the pre-trained model. However, as explained in Sec. 1, fine-tuning the pre-trained model can cause serious CF for CL. The CL system PCL [18], which uses this approach, has the pre-trained model frozen to avoid forgetting. But as we will see in Sec. 5.4, this is not the best choice. CTR inserts the proposed CL-plugin in two locations in BERT, i.e., in each transformer layer of BERT. We will also see in Sec. 5.4 that inserting only one CL-plugin in one location is sub-optimal. Figure 1 gives the CTR architecture and we can see the two CL-plugins are added into BERT. In learning, only the two CL-plugins and the classification heads are trained. The components of the original pre-trained BERT are fixed.

**Continual learning plug-in (CL-plugin).** CL-plugin employs a capsule network (CapsNet) [15, 50] like architecture. In the classic neural network, a neuron outputs a scalar, real-valued activation as a feature detector. CapsNets replaces that with a vector-output capsule to preserve additional information. A simple CapsNet consists of two capsule layers. The first layer stores low-level feature maps, and the second layer generates the classification probability with each capsule corresponding to one class. CapsNet uses a *dynamic routing* algorithm to make each lower-level capsule to send its output to a similar (or "agreed", computed by dot product) higher-level capsule. This property can already be used to group similar tasks and their shareable features to produce a CL system (see the ablation study in Sec. 5.4). One of the key ideas of CL-plugin (see Figure 2(A)) is a *transfer capsule layer* with a new *transfer routing* algorithm to explicitly identify transferable features/knowledge from previous tasks to transfer to the new task. Additionally, transfer routing avoids the need for hyper-parameter tuning on the number of iterations of dynamic routing [50] to update the agreements.

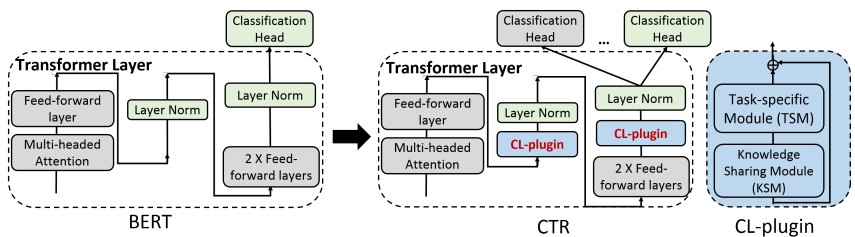

Figure 1: Architecture of BERT **(left)** and the proposed system CTR **(right)**, which inserts two CL-plugins in BERT. Each CL-plugin module (**far right**) has two sub-modules and a skip connection: knowledge sharing sub-module (KSM) and task-specific sub-module (TSM).

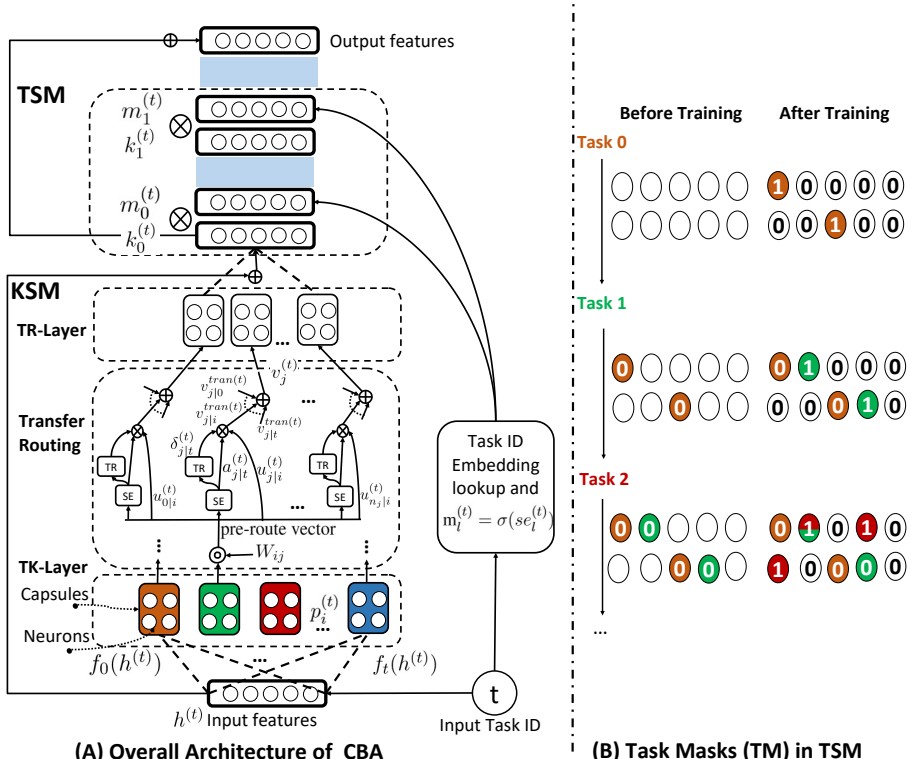

Figure 2: **(A)** CL-plugin Architecture. **(B)** Illustration of task masking. Cells/neurons in brown, green and red are respectively used by tasks 0, 1 and 2. Neurons with two colors are used by two tasks

## 4 Continual Learning Plug-in (CL-plugin)

The architecture of our *continual learning plug-in* (CL-plugin) is shown in Figure 2(A). CL-plugin takes two inputs: (1) hidden states $h^{(t)}$ from the feed-forward layer inside a transformer layer and (2) task ID $t$, which is required by task continual learning (Task-CL). The outputs are hidden states with features suitable for the $t$-th task for classification. Inside CL-plugin, there are two modules: (1) *knowledge sharing module* (KSM) for identifying and transferring the shareable knowledge from similar previous tasks to the new task $t$, and (2) *task specific module* (TSM) for learning task specific neurons and their masks (which can protect the neurons from being updated by future tasks to deal with CF). Since TSM is differentiable, the whole system CTR can be trained end-to-end.

### 4.1 Knowledge Sharing Module (KSM)

KSM achieves knowledge transfer among similar tasks via a *task capsule layer* (TK-Layer), a *transfer capsule layer* (TR-Layer), and a *transfer routing* mechanism.

### 4.1.1 Task Capsule Layer (TK-Layer)

Each capsule in the TK-Layer represents a task, and it prepares the low-level features derived from each task (Figure 2(A)). As such, a capsule is added to the TK-Layer for each new task. This incremental growth is efficient and easy because these capsules are discrete and do not share parameters. Also, each capsule is simply a 2-layer fully connected network with a small number of parameters. Let $h^{(t)} \in \mathbb{R}^{d_t \times d_e}$ be the input of CL-plugin, where $d_t$ is the number of tokens, $d_e$ the number of dimensions, and $t$ is the current task. In the TK-Layer, we have one capsule for each task. Assume we have learned $t$ tasks so far. The capsule for the $i$-th ($i \leq t$) task is

$$p_i^{(t)} = f_i(h^{(t)}), \tag{1}$$

where $f_i(\cdot) = \text{MLP}_i(\cdot)$ denotes a 2-layer fully-connected network.

### 4.1.2 Transfer Routing and Transfer Capsule Layer

Each capsule in the *transfer capsule layer* (TR-Layer) represents the transferable representation extracted from TK-Layer. As shown in Figure 2(A), *transfer routing* between the lower-level capsules in TK-Layer and high-level capsules in TR-Layer has three components: *pre-route vector generator* (*PVG*), *similarity estimator (SE)* and *task router (TR)*. Given the task capsules in the TK-Layer, we first transform the feature through a trainable weight matrix. We call the output of this transformation the *pre-route vector*. Each SE estimates the similarity between a previous task and the current task using the pre-route vector, resulting in a similarity score for each higher-level capsule. Additionally, each SE is augmented by a TR module, a differentiable task router acting as a gate. This router estimates a binary signal that decides whether to connect or disconnect the current route between the two consecutive capsule layers (i.e. TK-Layer and TR-Layer in CL-plugin). The binary signal estimated by TR can be seen as a differentiable binary attention. Conceptually, each SE and TR pair together learns the connectivity between capsules in a stochastic and differentiable manner, which can be seen as a task similarity-based connectivity search mechanism. This transfer routing identifies the shared features/knowledge from multiple task capsules and helps knowledge transfer across similar tasks. Next, we discuss the *pre-route vector generator*, *similarity estimator* and *task router*.

**Pre-route Vector Generator (PVG).** We first turns each transfer capsule $p_i^{(t)}$ into a pre-route vector,

$$u_{j|i}^{(t)} = W_{ij} p_i^{(t)}, \tag{2}$$

where $W_{ij} \in \mathbb{R}^{d_s \times d_k}$ is the weight matrix, $d_s$ and $d_k$ are the dimensions of task capsule $i$ (also representing a task) and transfer capsule $j$, and $t$ is the current task. The number of transfer capsules $n_j$ is a hyperparameter detailed in Sec. 5.

**Similarity Estimator (SE).** Since tasks $i$ and $t$ are different, it is crucial to determine what in task $i$'s representation is transferable. Inspired by [67], we use a convolution layer and activation units to compare task $i$ with task $t$ to determine the transferable proportion from the previous task $i$. In SE, we compute the task similarity as follows:

$$q_{j|t}^{(t)} = \text{MaxPool}(\text{Relu}(u_{j|t}^{(t)} * W_q + b_q)), \tag{3}$$

$$a_{j|i}^{(t)} = \text{MaxPool}(\text{Relu}(u_{j|i}^{(t)} * W_a + f_a(q_{j|t}^{(t)}) + b_a)), \tag{4}$$

where $b_a, b_q \in \mathbb{R}$ are the bias, $W_a, W_q \in \mathbb{R}^{d_e \times d_w}$ are convolutions filters and $d_w$ is the windows size. We extract important features from the current task representation $u_{j|t}^{(t)}$ via the convolution network in Eq. (3). The MaxPool helps remove the insignificant features to generate a fixed-size vector with the size equal to the number of filters $n_w$. Similarly, we extract important features from the previous task $i$'s representation $u_{j|i}^{(t)}$. Using the important features for the current and previous tasks, we compute a similarity score between them in Eq. (4) with ReLU activation. Note $f_a$ is a 1-layer fully-connected network to match the dimensions. As a result, $a_{j|i}^{(t)}$ indicates how similar the representation of the $i$-th task is to the current task $t$. For those tasks with a very low $a_{j|i}^{(t)}$, their representations are less similar to the current task and thus has little transferable knowledge.

**Task Router (TR).** TR controls which previous task representation should flow to the next layer with the goal of letting only the transferable information to flow. Given the similarity $a_{j|i}^{(t)}$, TR estimates

a binary decision signal $\delta_{j|t}^{(t)} \in \{\texttt{0:disconnect, 1:connect}\}$. We first apply a convolution layer with 2 output channels and $1 \times 1$ kernel size to generate un-normalized decision value. To estimate the binary decision, we need to generate a decision chosen from the set of two mutually exclusive and exhaustive events (disconnect and connect). In our work, we adopt the Gumbel-Softmax [19] to help make the TR gate differentiable.

$$\delta_{j|i}^{(t)} = \text{Gumbel\_softmax}(a_{j|i}^{(t)} * W_\delta + b_\delta). \tag{5}$$

Given the similarity $a_{j|t}^{(t)}$, binary decision $\delta_{j|t}^{(t)}$ and the pre-route vector $u_{j|t}^{(t)}$, we compute the transferable representation $v_{j|i}^{\text{tran}(t)}$ and final output $v_j^{(t)}$ as follows:

$$v_{j|i}^{\text{tran}(t)} = a_{j|i}^{(t)} \otimes u_{j|i}^{(t)}, \qquad v_j^{(t)} = \sum_{\substack{i=1 \\ \delta_{ij}^{(t)}=1}}^{n+1} v_{j|i}^{\text{tran}(t)}. \tag{6}$$

This makes sure only task capsules for tasks that are salient or similar to the new task are used, and the others task capsules are ignored (and thus protected) to learn more general shareable knowledge. As many NLP applications have similar tasks, such learning of task-sharing features can be very important. Note that in backpropagation, only the similar tasks with connected gate ($\delta_{ij}^{(t)}$=1) are updated, encouraging backward knowledge transfer of similar tasks.

### 4.2 Task Specific Module (TSM)

We now discuss how to preserve task-specific knowledge of previous tasks to prevent forgetting (CF). To achieve this, we use task masks (Figure 2(B)). Specifically, we first detect the neurons used by each old task and then block off or mask out all the *used* neurons when learning a new task.

The task-specific module consists of differentiable layers (a 2-layer fully-connected network is used). Each layer's output is further applied with a task mask to indicate which neurons should be protected for that task to overcome CF and forbids gradient updates for those neurons during backpropagation for a new task. Those tasks with overlapping masks indicate some parameter sharing. Due to KSM, the features flowing into those overlapping neurons enable the related old tasks to also improve in learning the new task. Here we borrow the hard attention idea in [52] and leverage the task ID embedding to train the task mask. Further details can be found in Supplementary.

**Illustration.** The task masking process is illustrated in Figure 2(B), which shows the learning process of three tasks. Before training, those solid cells with a 0 are the neurons that have been used by some previous tasks and should be protected (masked). Those empty cells are free neurons (not used). After training, those solid cells with a 1 are neurons that are important for the current task, which will be used as masks in the future. Those solid cells with a 0 are masked as they are important for previous tasks. Those non-solid cells with a 0 are neurons that are not used so far.

Let us walk through the learning process of the three tasks. After training task 0, we obtain its useful neurons indicated by the 1 entries. Before training task 1, those useful neurons for task 0 are first masked (those previous 1's entries are turned to 0's). After training task 1, two neurons with 1 are used by the task. When task 2 arrives, all used neurons by tasks 0 and 1 are masked before training, i.e., their entries are set to 0. After training task 2, we see that tasks 2 and 1 have a shared neuron (the cell with two colors, red and green), which is used by both of tasks.

## 5 Experiments

We evaluate CTR using three applications. We follow the continual learning (CL) evaluation method given in [29]. CTR first learns a sequence of tasks. After a task is trained, the training data of the task is no longer accessible. After all tasks are learned, their task models are tested using their respective test sets. In training each task, we use its validation set to decide when to stop training.

### 5.1 Three Applications and Their Datasets

The first two applications (and datasets) are used to show the knowledge transferability of CTR because their tasks are similar and have shared knowledge. Catastrophic forgetting (CF) is not a

| Data source | Liu3domain | | | HL5domain | | | | | Ding9domain | | | | | | | | | SemEval14 | |
|---|---|---|---|---|---|---|---|---|---|---|---|---|---|---|---|---|---|---|---|
| Task/ domain | Speaker | Router | Computer | Nokia6610 | Nikon4300 | Creative | CanonG3 | ApexAD | CanonD500 | Canon100 | Diaper | Hitachi | Ipod | Linksys | MicroMP3 | Nokia6600 | Norton | Restaurant | Laptop |
| Train | 352 | 245 | 283 | 271 | 162 | 677 | 228 | 343 | 118 | 175 | 191 | 212 | 153 | 176 | 484 | 362 | 194 | 3452 | 2163 |
| Val. | 44 | 31 | 35 | 34 | 20 | 85 | 29 | 43 | 15 | 22 | 24 | 26 | 19 | 22 | 61 | 45 | 24 | 150 | 150 |
| Test | 44 | 31 | 36 | 34 | 21 | 85 | 29 | 43 | 15 | 22 | 24 | 27 | 20 | 23 | 61 | 46 | 25 | 1120 | 638 |

Table 1: Statistics of datasets for ASC. The datasets statistics for DSC and 20News have been described in the text. More detailed data statistics are given in *Supplementary*.

major concern for them. The third application (and dataset) is mainly used to test CTR's ability to overcome CF as its tasks are very different and have little shared knowledge to transfer.

**1. Document Sentiment Classification (DSC).** This application is to classify each full product review into one of the two opinion classes (*positive* and *negative*). The training data of each task consists of reviews of a particular type of product. We adopt the text classification formulation in [8], where a [CLS] token is used to predict the opinion polarity.

We employ a set of 10 DSC datasets (reviews of 10 products) to produce sequences of tasks. The products are Sports, Toys, Tools, Video, Pet, Musical, Movies, Garden, Offices, and Kindle [22]. Two experiments are conducted: (1) using small data in each task: 100 positive and 100 negative training reviews per task; (2) using the full data in each task: 2500 positive and 2500 negative training reviews per task [22]. (1) is more useful in practice because labeling a large number of examples is very costly. The same validation reviews (250 positive and 250 negative) and the same test reviews (250 positive and 250 negative) are used in both experiments.

**2. Aspect Sentiment Classification (ASC).** It classifies a review sentence on the aspect-level sentiment (one of *positive*, *negative*, and *neutral*). For example, the sentence "*The picture is great but the sound is lousy*" about a TV expresses a *positive* opinion about the aspect "picture" and a *negative* opinion about the aspect "sound." We adopt the ASC formulation in [65], where the aspect term and sentence are concatenated via [SEP] in BERT. The opinion is predicted on top of the [CLS] token.

We employ a set of 19 ASC datasets (review sentences of 19 products) to produce sequences of tasks. Each dataset represents a task. The datasets are from 4 sources: (1) **HL5Domains** [17] with reviews of 5 products; (2) **Liu3Domains** [32] with reviews of 3 products; (3) **Ding9Domains** [9] with reviews of 9 products; and (4) **SemEval14** with reviews of 2 products - SemEval 2014 Task 4 for laptop and restaurant. For (1), (2) and (3), we use about 10% of the original data as the validate data, another about 10% of the original data as the testing data. For (4), we use 150 examples from the training set for validation. To be consistent with existing research [59], sentences with conflict opinions about a single aspect are not used. Statistics of the 19 datasets are given in Table 1.

**3. Text classification using 20News data**. This dataset [28] has 20 classes and each class has about 1000 documents. The data split for train/validation/test is 1600/200/200. We created 10 tasks, 2 classes per task. Since this is topic-based text classification data, the classes are very different and have little shared knowledge. As mentioned above, this application (and dataset) is mainly used to show CTR's ability to overcome forgetting.

### 5.2 Baselines

We setup 38 baselines, including both *standalone* and *continual learning* methods.

**Multitask learning (MTL**: Results of multitask learning is considered the upper-bound of those of continual learning. Here MTL fine-tunes BERT.

**Standalone (SDL) Baselines**: The SDL setting builds a model for each task independently using a separate network. It clearly has no knowledge transfer or forgetting. We have 4 baselines under SDL, **BERT**, **BERT (Frozen)**, **Adapter-BERT** and **W2V** (word2vec embeddings). For **BERT**, we use trainable BERT, i.e., fine-tuning. **BERT (Frozen)** uses frozen BERT with a trainable CNN text classification network [26] on top of it.; **Adapter-BERT** adapts the BERT as in [16], where only the adapter blocks are trainable; **W2V** uses embeddings trained on the Amazon review data in [66] using FastText [11]. For ASC, we adopt the ASC classification network in [67], which takes aspect term and review sentence as input. For DSC and 20News, we adopt the classification network in [8].

**Continual Learning (CL) Baselines**. CL setting includes 4 baselines with *no forgetting handling* (**NFH**) (corresponding to the 4 standalone baselines), and 25 baselines from 9 state-of-the art *task continual learning* (Task-CL) methods that deal with forgetting (CF). NFH baselines learn the tasks one by one with no awareness of forgetting/transfer.

The 12 state-of-the-art CL systems are: **KAN** [22] and **SRK** [37] are Task-CL methods for document sentiment classification. The rest were designed initially for image classification. Therefore, we replace their original MLP or CNN image classification network with CNN for text classification [26]. **HAT** [52] is one of the best Task-CL methods with almost no forgetting. **CAT** [21] is similar to HAT but can work with a mixed sequence. **UCL** [1] is a recent Task-CL method. **EWC** [27] is a popular regularization-based class continual learning (Class-CL) method, which was adapted for Task-CL by only training on the corresponding head of the specific task ID during training and only considering the corresponding head's prediction during testing. **OWM** [68] is also a Class-CL method, which we also adapt to Task-CL. **L2** [27] is a classic regularization based Class-CL method, which we adapt to Task-CL. **A-GEM** [4] is an efficient version of the replay Task-CL method GEM [36], which penalizes the previous task loss from being increased. **DER++** [2] is a recent replay method using distillation to regularize the new task training and it can function as a Task-CL method. **B-CL** [24] is similar to CTR but uses dynamic routing for knowledge transfer and it performs markedly worse than CTR. **LAMOL** [58] is a pseudo-replay method based on GPT-2.

From the 10 systems, we created 10 baselines using **W2V** embeddings with the aspect term added before the sentence so that the Task-CL methods can take both aspect and the review sentence for ASC; 7 baselines using **Adapter-BERT** (SRK, KAN and CAT's architectures cannot work with adapters); and 10 baselines using **BERT (Frozen)** (which replaces W2V embeddings). The BERT formulation in Sec. 3 naturally takes both aspect and review sentence in the ASC case.

## 5.3 Hyper-parameters

Unless otherwise stated, the same hyper-parameters are used in experiments for ASC, DSC and 20News datasets. For the knowledge sharing module (KSM), we employ 2 layers of fully connected network with dimensions 768 in the TK-Layer. We employ 3 transfer capsules. We experimented with 2 to 15 capsules and selected 3 based on the validation data accuracy. For the task specific module (TSM), we use 2000 dimensions as the final and the hidden layers of the TSM. The task ID embeddings have 2000 dimensions. A fully connected layer with softmax output is used as the classification heads in the last layer of the BERT, together with the categorical cross-entropy loss. dropout of 0.5 between fully connected layers. The training of BERT, Adapter-BERT and CTR follows that of [65]. We adopt BERT$_{BASE}$ (uncased). The maximum input length is set to 128 which is long enough for both ASC and DSC. We use Adam optimizer and set the learning rate to 3e-5. We use 10 epochs for SemEval datasets and 30 epochs for the other datasets in the ASC application. For DSC, we use 20 epochs. For 20News, we use 10 epochs. These are selected based on the validation results. The batch size is set to 32 for all cases. For all the other CL baselines, we use the code provided by their authors and adapt them for text classification. We also adopt their original parameters, including learning rate, early stopping, and batch size.

## 5.4 Evaluation Results and Analysis

Since the order of the tasks in a sequence may impact the final results, we randomly sampled 5 task sequences and averaged their results. We compute both accuracy and Macro-F1, where Macro-F1 is the primary metric as the imbalanced classes (in ASC) introduce biases on accuracy. Table 2 gives the average results of all the 19 tasks (or datasets) for ASC and 10 tasks (or datasets) for DSC and 20News over the 5 random task sequences.

**Overall Performance.** Table 2 shows that CTR clearly outperforms all baselines.

(1). For the standalone (SDL) baselines, BERT (with fine-tuning) and Adapter-BERT perform similarly. W2V and BERT (Frozen) are poorer, which is understandable.

(2). Comparing SDL (standalone learning) and NFH (continual learning with no forgetting handling), we see NFH is much better than SDL for W2V and BERT (Frozen). This indicates ASC and DSC tasks have similarities and thus shared knowledge. Catastrophic forgetting (CF) is not a major issue for W2V and BERT (Frozen). However, for 20News, NFH variants have serious CF. NFH with BERT (fine-tuning) is much worse than SDL and Adapter-BERT, which we explained in Introduction.

| Scenarios | Category | Model | ASC | | DSC (small) | | DSC (full) | | 20News | | 20News (FR) | |
|---|---|---|---|---|---|---|---|---|---|---|---|---|
| | | | Acc. | MF1 | Acc. | MF1 | Acc. | MF1 | Acc. | MF1 | Acc. | MF1 |
| Non-continual Learning (SDL) | BERT | MTL | 91.91 | 88.11 | 85.05 | 84.03 | 89.77 | 89.28 | 96.77 | 96.77 | — | — |
| | BERT | SDL | 85.84 | 76.35 | 78.04 | 74.17 | 87.84 | 86.80 | 96.49 | 96.48 | — | — |
| | BERT (Frozen) | SDL | 78.14 | 58.13 | 73.88 | 67.97 | 85.34 | 80.17 | 96.49 | 96.48 | — | — |
| | Adapter-BERT | SDL | 85.96 | 78.07 | 76.31 | 71.04 | 88.30 | 87.31 | 96.20 | 96.19 | — | — |
| | W2V | SDL | 77.01 | 51.89 | 62.06 | 53.80 | 69.57 | 65.51 | 94.72 | 94.72 | — | — |
| Continual Learning (CL) | BERT | NFH | 49.60 | 43.08 | 73.87 | 69.44 | 73.08 | 71.81 | 52.50 | 39.22 | 24.29 | 30.52 |
| | BERT (Frozen) | NFH | 85.51 | 76.64 | 83.12 | 79.23 | 61.88 | 45.79 | 83.28 | 81.81 | 8.76 | 9.73 |
| | Adapter-BERT | NFH | 54.03 | 44.81 | 63.76 | 53.95 | 64.94 | 63.40 | 68.29 | 61.70 | 30.59 | 3.79 |
| | W2V | NFH | 82.69 | 73.56 | 65.16 | 57.48 | 70.40 | 68.03 | 90.74 | 90.59 | 4.30 | 4.47 |
| | BERT (frozen) | L2 | 56.04 | 38.40 | 59.17 | 48.39 | 69.80 | 62.63 | 72.14 | 65.39 | 24.57 | 32.05 |
| | | A-GEM | 86.06 | 78.44 | 59.33 | 45.94 | 70.67 | 61.77 | 93.31 | 92.95 | 4.09 | 4.48 |
| | | DER++ | 84.27 | 75.08 | 72.29 | 66.28 | 86.70 | 85.46 | 60.44 | 49.67 | 10.54 | 12.16 |
| | | KAN | 85.49 | 77.38 | 77.27 | 72.34 | 82.32 | 81.23 | 73.07 | 69.97 | 15.52 | 18.87 |
| | | SRK | 84.76 | 78.52 | 78.58 | 76.03 | 83.99 | 82.66 | 79.64 | 77.89 | 12.06 | 13.97 |
| | | EWC | 86.37 | 74.52 | 82.38 | 78.41 | 72.77 | 65.76 | 80.26 | 78.60 | 3.50 | 3.03 |
| | | UCL | 83.89 | 74.82 | 80.12 | 74.13 | 74.76 | 69.48 | 94.65 | 94.63 | 0.48 | 0.48 |
| | | OWM | 87.02 | 79.31 | 58.07 | 42.63 | 86.30 | 85.36 | 84.54 | 82.73 | 13.80 | 15.81 |
| | | HAT | 86.74 | 78.16 | 79.48 | 72.78 | 87.29 | 86.14 | 93.51 | 92.93 | 2.26 | 2.89 |
| | | CAT | 83.68 | 68.64 | 67.41 | 56.22 | 87.34 | 86.51 | 95.17 | 95.16 | 0.80 | 0.81 |
| | Adapter-BERT | L2 | 63.97 | 52.43 | 67.26 | 62.76 | 73.03 | 71.50 | 69.56 | 65.50 | 23.12 | 27.39 |
| | | A-GEM | 45.88 | 28.21 | 62.89 | 55.96 | 71.22 | 69.94 | 60.29 | 50.40 | 40.22 | 51.20 |
| | | DER++ | 47.63 | 35.54 | 70.52 | 63.56 | 59.67 | 57.82 | 58.95 | 49.58 | 36.39 | 45.30 |
| | | EWC | 56.30 | 49.58 | 58.23 | 51.03 | 62.69 | 61.51 | 61.86 | 53.94 | 37.79 | 46.58 |
| | | UCL | 64.46 | 36.64 | 48.30 | 32.07 | 57.06 | 55.86 | 51.75 | 36.06 | 4.70 | 6.60 |
| | | OWM | 72.99 | 66.51 | 73.97 | 71.96 | 85.46 | 84.57 | 71.10 | 66.25 | 27.38 | 32.76 |
| | | HAT | 86.14 | 78.52 | 80.83 | 78.41 | 88.00 | 87.26 | 95.22 | 95.21 | 0.33 | 0.34 |
| | W2V | L2 | 60.36 | 39.13 | 54.34 | 43.19 | 57.71 | 48.00 | 59.54 | 54.40 | 7.83 | 11.89 |
| | | A-GEM | 81.33 | 63.35 | 69.80 | 60.07 | 77.67 | 70.75 | 90.72 | 90.60 | 3.94 | 4.08 |
| | | DER++ | 83.27 | 69.93 | 77.51 | 73.13 | 74.79 | 66.68 | 89.28 | 89.19 | 4.32 | 4.42 |
| | | KAN | 72.06 | 40.01 | 57.13 | 43.75 | 69.35 | 64.78 | 57.92 | 51.65 | 20.98 | 27.02 |
| | | SRK | 71.01 | 39.63 | 64.47 | 55.93 | 69.65 | 65.25 | 61.07 | 58.47 | 7.26 | 8.81 |
| | | EWC | 84.16 | 72.29 | 64.82 | 57.20 | 70.00 | 65.11 | 91.86 | 91.80 | 2.64 | 2.71 |
| | | UCL | 84.41 | 75.99 | 56.23 | 41.34 | 70.56 | 67.01 | 90.61 | 90.46 | 4.53 | 4.70 |
| | | OWM | 82.70 | 71.18 | 53.40 | 38.44 | 67.15 | 65.42 | 71.97 | 68.75 | 24.00 | 27.56 |
| | | HAT | 80.83 | 63.63 | 62.57 | 50.83 | 69.75 | 65.44 | 67.73 | 64.43 | 26.04 | 29.70 |
| | | CAT | 76.28 | 54.65 | 55.19 | 35.28 | 79.58 | 75.99 | 70.38 | 68.04 | 24.37 | 26.95 |
| | B-CL | | 88.29 | 81.40 | 82.01 | 80.63 | 79.76 | 76.51 | 95.07 | 95.04 | 0.58 | 0.59 |
| | LAMOL | | 88.91 | 80.59 | 89.12 | 86.58 | 92.11 | 91.72 | 66.13 | 45.74 | 20.03 | 16.60 |
| | CTR (forward) | | 87.89 | 80.25 | 83.75 | 82.55 | 89.86 | 89.16 | 95.63 | 95.62 | — | — |
| | CTR | | 89.47 | 83.62 | 84.34 | 83.29 | 89.31 | 88.75 | 95.25 | 95.23 | 0.42 | 0.43 |

Table 2: Accuracy (Acc.) and Macro-F1 (MF1), averaged over 5 random sequences. The architectures of SRK, KAN and CAT cannot work with Adapter-BERT. "—" means not applicable. Due to the limited space, *standard deviation*, *running time* and *network size* are given in Supplementary.

(3). Unlike BERT and Adapter-BERT, CTR does very well in both forgetting avoidance and knowledge transfer (outperforming all baselines). For baselines, EWC, UCL, OWM and HAT, although they perform better than NFH, they are significantly poorer than CTR as they don't have methods to encourage knowledge transfer for ASC and DSC. KAN and SRK do knowledge transfer but they are weaker than many other CL methods. They perform very poorly for 20News as they have limited ability to overcome CF. CAT works well with large datasets, but is weak for small datasets.

(4). CTR's improvements over SDL variants for DSC (large) is less than for DCS (small). This is understandable because when the training data is large, learning a separate model is already good, and knowledge transfer is less important.

(5). Compared with the SDL results, we can see that CTR has the least forgetting on 20News.

(6). Compared to B-CL, CTR is markedly better in knowledge transfer. The forgetting rates (FR) of B-CL and CTR are both low. The comparison is in fact the same as comparing dynamics routing and transfer routing. We can see that the proposed transfer routing is dramatically better than dynamic routing for knowledge transfer. Additionally, transfer routing eliminates the need for hyper-parameter tuning on the number of iterations of dynamic routing [50] to update the agreements.

(7). CTR outperforms LAMOL in ASC and 20News even with the less powerful BERT model that CTR adopts. LAMOL outperforms BERT-based MTL in DSC. This may be because LAMOL is based on GPT-2, which is known to be more powerful than BERT (also shown in the LAMOL paper). For 20News, since its tasks are very different/dissimilar, there is little shared knowledge. Dealing with CF is the main issue. LAMOL has serious forgetting as its FR values show.

(8). The results of MTL (the upper bound) are only slightly better than CTR, which again shows that CTR is highly effective in overcoming forgetting and encouraging knowledge transfer.

| Model | ASC | | DSC (small) | | DSC (full) | | 20News | |
|---|---|---|---|---|---|---|---|---|
| | Acc. | MF1 | Acc. | MF1 | Acc. | MF1 | Acc. | MF1 |
| CTR (-KSM;-TSM) | 0.5403 | 0.4481 | 0.6376 | 0.5395 | 0.6494 | 0.6340 | 0.6829 | 0.6170 |
| CTR (-TSM) | 0.8312 | 0.7107 | 0.7085 | 0.6759 | 0.8545 | 0.8380 | 0.8275 | 0.8064 |
| CTR (-KSM) | 0.8614 | 0.7852 | 0.8083 | 0.7841 | 0.8800 | 0.8726 | 0.9522 | 0.9521 |
| CTR (-TR/KSM) | 0.8819 | 0.8155 | 0.8244 | 0.8119 | 0.8831 | 0.8762 | 0.9476 | 0.9469 |
| CTR (dynamic routing) (B-CL) | 0.8829 | 0.8140 | 0.8201 | 0.8063 | 0.7976 | 0.7651 | 0.9507 | 0.9504 |
| CTR (on top) | 0.8135 | 0.6390 | 0.7301 | 0.6875 | 0.8266 | 0.8105 | 0.8927 | 0.8920 |
| CTR (after the first FF layer) | 0.8741 | 0.8014 | 0.8300 | 0.8183 | 0.8699 | 0.8596 | 0.9381 | 0.9373 |
| CTR (after the other two FF layers) | 0.8662 | 0.7863 | 0.8269 | 0.8161 | 0.8714 | 0.8612 | 0.9339 | 0.9316 |
| **CTR** | **0.8947** | **0.8362** | **0.8434** | **0.8329** | **0.8931** | **0.8875** | **0.9525** | **0.9523** |

Table 3: Ablation experiment results.

**Effectiveness of Knowledge Transfer.** Forward transfer is defined as the forward performance (**CTR(forward)** in Table 2) subtracting the standalone (SDL) result. CTR(forward) is the test accuracy or MF1 of each task when it was first learned. Backward transfer is defined as the difference between the backward performance (**CTR** in Table 2, the final result after all tasks are learned) and the forward performance. The average results of CTR (forward) and CTR are given in Table 2. We observe that forward transfer of CTR is highly effective for the three datasets with similar tasks. For DSC, the less the data, the more effective is the transfer, which is reasonable. Backward transfer improves the accuracy and MF1 of ASC and DSC (small). For DSC (full), it is slightly weaker and for 20News, it is also slightly weaker due to a very small amount of forgetting, but the less than 0.0055 CF is negligible. Note that in [36, 46], forward transfer is measured by comparing the test results of the new task on the current learned network and a random initialized network before/without training the new task (like zero-shot). This method indicates whether the learned network contains some useful knowledge for the new task. However, it does not tell how much forward transfer actually happens after learning the new task, which is more important and is what our method measures.

**Overcoming Forgetting.** To validate CTR's effectiveness in dealing forgetting with a sequence of dissimilar tasks, we compute the *Forgetting Rate* [34], $FR = \frac{1}{t-1} \sum_{i=1}^{t-1} A_{i,i} - A_{t,i}$, where $A_{i,i}$ is the forward accuracy of task $i$ and $A_{t,i}$ is the accuracy of task $i$ after training the last task $t$. We average over all tasks except the last one because the last task obviously has no forgetting. We report the forgetting rate FR (averaged over 5 random task sequences) for the 20News data on the two evaluation metrics in the last two columns of Table 2 (the other two datasets are mainly for knowledge transfer). CTR has very low FR values which indicate very little forgetting.

**Ablation Study.** The results of ablation experiments are in Table 3. "-KSM;-TSM" means that the knowledge sharing module (KSM) and the task specific module (TSM) are not used, and we simply deploy the Adapter-BERT. "-KSM" means that the knowledge sharing module (KSM) is not used. "-TSM" means that the task specific module (TSM) is not used. "-TR/KSM" means that the task router (TR) in KSM is not used. We directly send the transferable representation $v_{j|i}^{tran(t)}$ to the next layer. "dynamic routing" means that we replace our transfer routing (-(SE&TR)/KSM) with dynamic routing [50], which is one of the most popular routing algorithms in capsule networks. "on top" means adding a CL-plugin on top of BERT. "after the first FF layer" means adding only one CL-plugin there in BERT. "after the other two FF layers" means adding only one CL-plugin module there in BERT. Table 3 shows that the full CTR system gives the best results, indicating every component contributes to the model and other options of adding CL-plugins are all poorer.

## 6 Conclusion

This paper studied task continual learning (Task-CL) using the pre-trained model BERT to achieve both CF presentation and knowledge transfer. It proposed a novel technique called CTR to leverage the pre-trained BERT for CL. The key component of CTR is the CL-plugin inserted in BERT. A CL-plugin is a capsule network with a new transfer routing mechanism to encourage knowledge transfer among tasks and also to isolate task-specific knowledge to avoid forgetting. Experimental results using three NLP applications showed that CTR markedly improves the performance of both the new task and the old tasks via knowledge transfer and is also effective at overcoming catastrophic forgetting. One limitation of our work is the efficiency due to the use of capsules. Capsules try to represent a group of neurons in a vector reflecting properties of an entity. In NLP, an entity is a sentence/document which contains many tokens (e.g., 128) and features (e.g. 768 in BERT$_{BASE}$). Grouping them makes the capsule very large (e.g., $128 \times 768$), which slows down training.

## Acknowledgments

This work was supported in part by two National Science Foundation (NSF) grants (IIS-1910424 and IIS-1838770), a DARPA Contract HR001120C0023, and a Northrop Grumman research gift.

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
