# Supplemental Materials for Achieving Forgetting Prevention and KnowledgeTransfer in Continual Learning

**Zixuan Ke[1], Bing Liu[1], Nianzu Ma[1], Hu Xu[2] and Lei Shu[3]***

[1]Department of Computer Science, University of Illinois at Chicago
[2]Facebook AI Research
[3]Amazon AWS AI
[1]{zke4,liub,nma4}@uic.edu
[2]huxu@fb.com
[3]shulindt@gmail.com

## 1   Task Masks (TM) in Task Specific Module (TSM)

In this section, we detail the task mask (TM) training. TMs (Figure 2(B) in the main paper) are used to prevent *catastrophic forgetting* (CF), i.e., to protect the task specific knowledge of previous tasks. Specifically, we first detect the neurons used by each old task, and then block off or mask out all the *used* neurons when learning a new task.

The task specific module (TSM) consists of differentiable layers (CBA uses a 2-layer fully-connected network). Each layer's output is further applied with a task mask to indicate which neurons should be protected for that task to overcome CF and forbids gradient updates for those neurons during backpropagation for a new task. Those tasks with overlapping masks indicate some parameter sharing. Due to KSM, the features flowing into those overlapping neurons enable the related old tasks to also improve in learning the new task.

**Task Masks.** Given the transfer capsule $v_j^{(t)}$, TSM maps them into input $k_l^{(t)}$ via a fully-connected network, where $l$ is the $l$-th layer in TSM. A task mask (a "soft" binary mask) $m_l^{(t)}$ is trained for each task $t$ at each layer $l$ in TSM during training task $t$'s classifier, indicating the neurons that are important for the task. Here we borrow the hard attention idea in [1] and leverage the task ID embedding to the train the task mask.

For a task ID $t$, its embedding $e_l^{(t)}$ consists of differentiable deterministic parameters that can be learned together with other parts of the network. It is trained for each layer in TSM. To generate the task mask $\mathrm{m}_l^{(t)}$ from $e_l^{(t)}$, *Sigmoid* is used as a pseudo-gate function and a positive scaling hyper-parameter $s$ is applied to help training. The $m_l^{(t)}$ is computed as follows:

$$m_l^{(t)} = \sigma(se_l^{(t)}). \tag{1}$$

Note that the neurons in $m_l^{(t)}$ may overlap with those in other $m_l^{(i_{\text{prev}})}$'s from previous tasks showing some shared knowledge. Given the output of each layer in TSM, $k_l^{(t)}$, we element-wise multiply $k_l^{(t)} \otimes m_l^{(t)}$. The masked output of the last layer $k^{(t)}$ is fed to the next layer of the BERT with a skip-connection (Figure 1 in the main paper). After learning task $t$, the final $m_l^{(t)}$ is saved and added to the set $\{m_l^{(t)}\}$.

---

*Work was done prior to joining Amazon.

35th Conference on Neural Information Processing Systems (NeurIPS 2021).

**Training.** For each past task $i_{\text{prev}} \in \mathcal{T}_{\text{prev}}$, its mask $m_l^{(i_{\text{prev}})}$ indicates which neurons are used by that task and need to be protected. In learning task $t$, $m_l^{(i_{\text{prev}})}$ is used to set the gradient $g_l^{(t)}$ on *all* used neurons of the layer $l$ in TSM to 0. Before modifying the gradient, we first accumulate all used neurons by all previous tasks' masks. Since $m_l^{(i_{\text{prev}})}$ is binary, we use max-pooling to achieve the accumulation:

$$m_l^{(t_{\text{ac}})} = \text{MaxPool}(\{m_l^{(i_{\text{prev}})}\}). \tag{2}$$

The term $m_l^{(t_{\text{ac}})}$ is applied to the gradient:

$$g_l^{'(t)} = g_l^{(t)} \otimes (1 - m_l^{(t_{\text{ac}})}). \tag{3}$$

Those gradients corresponding to the 1 entries in $m_l^{(t_{\text{ac}})}$ are set to 0 while the others remain unchanged. In this way, neurons in an old task are protected. Note that we expand (copy) the vector $m_l^{(t_{\text{ac}})}$ to match the dimensions of $g_l^{(t)}$.

Though the idea is intuitive, $e_l^{(t)}$ is not easy to train. To make the learning of $e_l^{(t)}$ easier and more stable, an annealing strategy is applied. That is, $s$ is annealed during training, inducing a gradient flow and set $s = s_{\max}$ during testing. Eq. 1 approximates a unit step function as the mask, with $m_l^{(t)} \to \{0, 1\}$ when $s \to \infty$. A training epoch starts with all neurons being equally active, which are progressively polarized within the epoch. Specifically, $s$ is annealed as follows:

$$s = \frac{1}{s_{\max}} + (s_{\max} - \frac{1}{s_{\max}})\frac{b-1}{B-1}, \tag{4}$$

where $b$ is the batch index and $B$ is the total number of batches in an epoch.

Let us walk through the learning process of the three tasks in Figure 2(B) in the main paper. After training task 0, we obtain its useful neurons indicated by the 1 entries. Before training task 1, those useful neurons for task 0 are first masked (those previous 1's entries are turned to 0's). After training task 1, two neurons with 1 are used by the task. When task 2 arrives, all used neurons by tasks 0 and 1 are masked before training, i.e., their entries are set to 0. After training task 2, we see that tasks 2 and 1 have a shared neuron (the cell with two colors, red and green), which is used by both of tasks.

## 2 Detailed Datasets Statistics

Since the datasets for the document sentiment classification (DSC) application (which is the same as a traditional classification problem) and the 20News dataset (which forms dissimilar task sequences and is used to show the forgetting avoidance ability) have already been described in Section 5.1 in the main paper, here we mainly focus on the datasets for aspect sentiment classification (ASC), which is more than a traditional classification problem because of the additional input of the aspect and the fact that in the same sentence different aspects can have different opinions. Table 1 in the main paper has provided the number of sentences or examples in each of the 19 datasets. However, no aspects or aspect opinions were provided. Here we provide them, as shown in Table 1.

## 3 Standard Deviations

We report the standard deviations (Table 2) of the accuracy (Acc.) and macro-F1 (MF1) results of CTR and the considered baselines over 5 runs with random seeds based on one random task sequence used in the paper. Note that this is different from Table 2 of the main paper where each result reported is the average result of 5 random task sequences as different task sequences can produce different results. We can see the results of CTR are stable. Some baselines can have quite large standard deviations using Adapter-BERT.

## 4 Execution Time and Number of Parameters

Table 3 reports the number of parameter (regardless of trainable or non-trainable), training execution times for different models. The execution time is computed as the average training time *per task*. Our experiments were run on GeForce GTX 2080 Ti with 11G GPU memory.

| Dataset | Tasks/Domains | Training | Validating | Testing |
|---|---|---|---|---|
| Liu3domain | Speaker | 233 S./352 A./287 P./65 N./0 Ne. | 30 S./44 A./35 P./9 N./0 Ne. | 38 S./44 A./40 P./4 N./0 Ne. |
| | Router | 200 S./245 A./142 P./103 N./0 Ne. | 24 S./31 A./19 P./12 N./0 Ne. | 22 S./31 A./24 P./7 N./0 Ne. |
| | Computer | 187 S./283 A./218 P./65 N./0 Ne. | 25 S./35 A./23 P./12 N./0 Ne. | 29 S./36 A./29 P./7 N./0 Ne. |
| HL5domain | Nokia6610 | 209 S./271 A./198 P./73 N./0 Ne. | 29 S./34 A./25 P./9 N./0 Ne. | 28 S./34 A./25 P./9 N./0 Ne. |
| | Nikon4300 | 131 S./162 A./135 P./27 N./0 Ne. | 15 S./20 A./18 P./2 N./0 Ne. | 15 S./21 A./19 P./2 N./0 Ne. |
| | Creative | 582 S./677 A./422 P./255 N./0 Ne. | 68 S./85 A./42 P./43 N./0 Ne. | 70 S./85 A./52 P./33 N./0 Ne. |
| | CanonG3 | 190 S./228 A./180 P./48 N./0 Ne. | 25 S./29 A./21 P./8 N./0 Ne. | 24 S./29 A./24 P./5 N./0 Ne. |
| | ApexAD | 281 S./343 A./146 P./197 N./0 Ne. | 35 S./43 A./16 P./27 N./0 Ne. | 28 S./43 A./31 P./12 N./0 Ne. |
| Ding9domain | CanonD500 | 103 S./118 A./96 P./22 N./0 Ne. | 11 S./15 A./14 P./1 N./0 Ne. | 13 S./15 A./11 P./4 N./0 Ne. |
| | Canon100 | 137 S./175 A./123 P./52 N./0 Ne. | 19 S./22 A./21 P./1 N./0 Ne. | 16 S./22 A./21 P./1 N./0 Ne. |
| | Diaper | 166 S./191 A./143 P./48 N./0 Ne. | 22 S./24 A./18 P./6 N./0 Ne. | 24 S./24 A./22 P./2 N./0 Ne. |
| | Hitachi | 152 S./212 A./153 P./59 N./0 Ne. | 23 S./26 A./19 P./7 N./0 Ne. | 23 S./27 A./14 P./13 N./0 Ne. |
| | Ipod | 124 S./153 A./101 P./52 N./0 Ne. | 18 S./19 A./14 P./5 N./0 Ne. | 19 S./20 A./15 P./5 N./0 Ne. |
| | Linksys | 152 S./176 A./128 P./48 N./0 Ne. | 19 S./22 A./13 P./9 N./0 Ne. | 20 S./23 A./16 P./7 N./0 Ne. |
| | MicroMP3 | 384 S./484 A./340 P./144 N./0 Ne. | 42 S./61 A./48 P./13 N./0 Ne. | 51 S./61 A./39 P./22 N./0 Ne. |
| | Nokia6600 | 298 S./362 A./244 P./118 N./0 Ne. | 26 S./45 A./32 P./13 N./0 Ne. | 39 S./46 A./30 P./16 N./0 Ne. |
| | Norton | 168 S./194 A./54 P./140 N./0 Ne. | 17 S./24 A./15 P./9 N./0 Ne. | 24 S./25 A./5 P./20 N./0 Ne. |
| SemEval14 | Rest | 1893 S./3452 A./2094 P./779 N./579 Ne. | 84 S./150 A./70 P./26 N./54 Ne. | 600 S./1120 A./728 P./196 N./196 Ne. |
| | Laptop | 1360 S./2163 A./930 P./800 N./433 Ne. | 98 S./150 A./57 P./66 N./27 Ne. | 411 S./638 A./341 P./128 N./169 Ne. |

Table 1: Statistics of the ASC datasets. **S.**: number of sentences; **A**: number of aspects; **P., N., and Ne.**: number aspects with positive, negative and neutral opinions, respectively. Note that the SemEval14 datasets have 3 classes of opinion polarities (positive, negative and neutral) while the others have only 2 classes (positive and negative) because in these other datasets, those sentences with neutral opinions were not annotated with aspects and thus cannot be used in *aspect* sentiment classification (ASC). That is why we have "0 Ne." for those datasets.

| Scenarios | Category | Model | ASC | | DSC (small) | | DSC (full) | | 20News | |
|---|---|---|---|---|---|---|---|---|---|---|
| | | | Acc. | MF1 | Acc. | MF1 | Acc. | MF1 | Acc. | MF1 |
| Non-continual Learning (SDL) | BERT | MTL | ±0.0073 | ±0.0088 | ±0.0111 | ±0.0117 | ±0.0034 | ±0.0037 | ±0.0049 | ±0.0049 |
| | BERT | SDL | ±0.0118 | ±0.0263 | ±0.0288 | ±0.0401 | ±0.0048 | ±0.0052 | ±0.0022 | ±0.0022 |
| | BERT (Frozen) | SDL | ±0.0171 | ±0.0265 | ±0.0019 | ±0.0027 | ±0.0042 | ±0.0063 | ±0.0044 | ±0.0044 |
| | Adapter-BERT | SDL | ±0.0175 | ±0.0154 | ±0.0081 | ±0.0150 | ±0.0053 | ±0.0060 | ±0.0048 | ±0.0048 |
| | W2V | SDL | ±0.0102 | ±0.0077 | ±0.0082 | ±0.0131 | ±0.0072 | ±0.0094 | ±0.0022 | ±0.0022 |
| Continual Learning (CL) | BERT | NFH | ±0.1051 | ±0.0492 | ±0.0274 | ±0.0363 | ±0.0736 | ±0.0701 | ±0.0518 | ±0.0508 |
| | BERT (Frozen) | NFH | ±0.0042 | ±0.0098 | ±0.0023 | ±0.0040 | ±0.0051 | ±0.0049 | ±0.0044 | ±0.0045 |
| | Adapter-BERT | NFH | ±0.0659 | ±0.0885 | ±0.0801 | ±0.0608 | ±0.0792 | ±0.0923 | ±0.0396 | ±0.0677 |
| | W2V | NFH | ±0.0133 | ±0.0325 | ±0.0064 | ±0.0206 | ±0.0203 | ±0.0337 | ±0.0132 | ±0.0146 |
| | BERT (frozen) | L2 | ±0.0618 | ±0.0405 | ±0.0320 | ±0.0134 | ±0.0358 | ±0.0731 | ±0.0161 | ±0.0230 |
| | | A-GEM | ±0.0078 | ±0.0142 | ±0.0036 | ±0.0036 | ±0.0037 | ±0.0042 | ±0.0037 | ±0.0037 |
| | | DER++ | ±0.0067 | ±0.0077 | ±0.0056 | ±0.0135 | ±0.0135 | ±0.0160 | ±0.0530 | ±0.0759 |
| | | KAN | ±0.0099 | ±0.0170 | ±0.0348 | ±0.0361 | ±0.0088 | ±0.0089 | ±0.0335 | ±0.0432 |
| | | SRK | ±0.0105 | ±0.0175 | ±0.0184 | ±0.0230 | ±0.0052 | ±0.0059 | ±0.0247 | ±0.0318 |
| | | EWC | ±0.0714 | ±0.0392 | ±0.0154 | ±0.0368 | ±0.0329 | ±0.0398 | ±0.0509 | ±0.0808 |
| | | UCL | ±0.0205 | ±0.0477 | ±0.0053 | ±0.0053 | ±0.0046 | ±0.0047 | ±0.0048 | ±0.0048 |
| | | OWM | ±0.0165 | ±0.0206 | ±0.0002 | ±0.0027 | ±0.0174 | ±0.0078 | ±0.0139 | ±0.0144 |
| | | HAT | ±0.0209 | ±0.0304 | ±0.0146 | ±0.0200 | ±0.0047 | ±0.0065 | ±0.0423 | ±0.0567 |
| | | CAT | ±0.0246 | ±0.0649 | ±0.0584 | ±0.1012 | ±0.0103 | ±0.0097 | ±0.0067 | ±0.0068 |
| | Adapter-BERT | L2 | ±0.0313 | ±0.0499 | ±0.0766 | ±0.1237 | ±0.0383 | ±0.0449 | ±0.0278 | ±0.0374 |
| | | A-GEM | ±0.0941 | ±0.0609 | ±0.0934 | ±0.1319 | ±0.0624 | ±0.0662 | ±0.0235 | ±0.0318 |
| | | DER++ | ±0.0853 | ±0.0712 | ±0.0813 | ±0.1195 | ±0.1005 | ±0.0963 | ±0.0984 | ±0.1161 |
| | | EWC | ±0.0943 | ±0.0991 | ±0.0610 | ±0.0831 | ±0.1209 | ±0.1215 | ±0.0409 | ±0.0616 |
| | | UCL | ±0.0731 | ±0.0341 | ±0.0436 | ±0.0203 | ±0.1017 | ±0.1022 | ±0.1322 | ±0.0890 |
| | | OWM | ±0.0347 | ±0.0419 | ±0.0381 | ±0.0344 | ±0.0046 | ±0.0044 | ±0.0316 | ±0.0461 |
| | | HAT | ±0.0058 | ±0.0091 | ±0.0042 | ±0.0119 | ±0.0197 | ±0.0205 | ±0.0037 | ±0.0205 |
| | W2V | L2 | ±0.0124 | ±0.0078 | ±0.0116 | ±0.0233 | ±0.0252 | ±0.0157 | ±0.0128 | ±0.0219 |
| | | A-GEM | ±0.0062 | ±0.0238 | ±0.0164 | ±0.0301 | ±0.0191 | ±0.0250 | ±0.0076 | ±0.0086 |
| | | DER++ | ±0.0059 | ±0.0130 | ±0.0249 | ±0.0415 | ±0.0163 | ±0.0219 | ±0.0140 | ±0.0148 |
| | | KAN | ±0.0111 | ±0.0044 | ±0.0083 | ±0.0162 | ±0.0473 | ±0.0302 | ±0.0067 | ±0.0065 |
| | | SRK | ±0.0074 | ±0.0029 | ±0.0161 | ±0.0162 | ±0.0031 | ±0.0057 | ±0.0123 | ±0.0151 |
| | | EWC | ±0.0264 | ±0.0581 | ±0.0478 | ±0.0952 | ±0.0155 | ±0.0208 | ±0.0412 | ±0.0424 |
| | | UCL | ±0.0148 | ±0.0110 | ±0.0056 | ±0.0131 | ±0.0209 | ±0.0313 | ±0.0113 | ±0.0121 |
| | | OWM | ±0.0258 | ±0.0299 | ±0.0228 | ±0.0348 | ±0.0167 | ±0.0194 | ±0.0196 | ±0.0249 |
| | | HAT | ±0.0194 | ±0.0203 | ±0.0192 | ±0.0220 | ±0.0484 | ±0.0553 | ±0.0422 | ±0.0700 |
| | | CAT | ±0.0114 | ±0.0278 | ±0.0001 | ±0.0002 | ±0.0182 | ±0.0242 | ±0.0251 | ±0.0317 |
| | B-CL | | ±0.0093 | ±0.0324 | ±0.0177 | ±0.0208 | ±0.0111 | ±0.0117 | ±0.0085 | ±0.0087 |
| | LAMOL | | ±0.0256 | ±0.0085 | ±0.0089 | ±0.0241 | ±0.0316 | ±0.0300 | ±0.0254 | ±0.0265 |
| | CTR | | ±0.0107 | ±0.0123 | ±0.0083 | ±0.0076 | ±0.0011 | ±0.0016 | ±0.0067 | ±0.0067 |

Table 2: Standard deviations of the accuracy (Acc.) and Macro-F1 (MF1) results of the proposed CTR model and the baselines on the four experiments.

# 5  Hyperparameter Search

We use grid search to find the best parameters based on the validation data performance. We search within {32, 64, 128} for batch size, within {140, 200, 300, 400} for $s_{max}$, within {300, 768, 2000} for dimension of Task Specific Module (TSM) and within {10, 20, 30, 40} for the number of BERT

| Scenarios | Category | Model | #Parameters (M) | Running time (min) | | | |
|---|---|---|---|---|---|---|---|
| | | | | ASC | DSC (small) | DSC (full) | 20News |
| Non-continual Learning (SDL) | BERT | MTL | 109.5 | 1.3 | 0.8 | 19.1 | 3.4 |
| | BERT | SDL | 109.5 | 2.1 | 1.7 | 23.8 | 4.9 |
| | BERT (Frozen) | SDL | 110.4 | 3.3 | 3.2 | 17.3 | 7.0 |
| | Adapter-BERT | SDL | 183.3 | 5.1 | 3.4 | 32.8 | 6.4 |
| | W2V | SDL | 6.7 | 0.7 | 0.2 | 0.6 | 0.5 |
| Continual Learning (CL) | BERT | NFH | 109.5 | 2.1 | 1.7 | 23.8 | 4.9 |
| | BERT (Frozen) | NFH | 110.4 | 3.3 | 3.2 | 17.3 | 7.0 |
| | Adapter-BERT | NFH | 183.3 | 5.1 | 3.4 | 32.8 | 6.4 |
| | W2V | NFH | 6.7 | 0.7 | 0.2 | 0.6 | 0.5 |
| | BERT (frozen) | L2 | 110.4 | 3.4 | 2.5 | 17.6 | 7.4 |
| | | A-GEM* | 110.4 | 3.3 | 3.2 | 17.3 | 7.0 |
| | | DER++* | 110.4 | 3.3 | 3.2 | 17.3 | 7.0 |
| | | KAN | 116.6 | 1.4 | 1.0 | 7.5 | 1.9 |
| | | SRK | 117.8 | 3.3 | 8.8 | 35.9 | 7.1 |
| | | EWC | 110.4 | 5.7 | 2.6 | 29.7 | 12.4 |
| | | UCL | 110.4 | 3.4 | 2.0 | 17.2 | 7.2 |
| | | OWM | 110.6 | 3.4 | 2.0 | 17.1 | 7.2 |
| | | HAT | 111.3 | 3.4 | 2.0 | 17.4 | 7.3 |
| | | CAT | 227.4 | 23.8 | 23.0 | 124.56 | 50.4 |
| | Adapter-BERT | L2 | 183.3 | 2.7 | 2.5 | 31.7 | 6.5 |
| | | A-GEM* | 183.3 | 5.1 | 3.4 | 32.8 | 6.4 |
| | | DER++* | 183.3 | 5.1 | 3.4 | 32.8 | 6.4 |
| | | EWC | 183.3 | 4.8 | 3.9 | 60.3 | 12.3 |
| | | UCL | 183.4 | 2.3 | 2.2 | 26.8 | 5.5 |
| | | OWM | 184.4 | 2.7 | 2.6 | 30.1 | 6.2 |
| | | HAT | 185.2 | 2.7 | 2.5 | 30.3 | 6.2 |
| | W2V | L2 | 6.2 | 8.2 | 0.2 | 0.6 | 0.5 |
| | | A-GEM* | 6.2 | 0.7 | 0.2 | 0.6 | 0.5 |
| | | DER++* | 6.2 | 0.7 | 0.2 | 0.6 | 0.5 |
| | | KAN | 7.0 | 0.1 | 0.1 | 0.2 | 0.1 |
| | | SRK | 7.2 | 2.4 | 2.8 | 3.1 | 4.2 |
| | | EWC | 6.2 | 1.2 | 0.4 | 3.0 | 1.4 |
| | | UCL | 6.2 | 0.7 | 0.3 | 0.7 | 0.5 |
| | | OWM | 6.4 | 0.7 | 0.2 | 0.8 | 0.5 |
| | | HAT | 6.4 | 0.8 | 0.3 | 1.0 | 0.6 |
| | | CAT | 24.5 | 5.0 | 1.4 | 4.5 | 3.6 |
| | B-CL | | 287.4 | 27.8 | 14.5 | 90.2 | 35.1 |
| | LAMOL | | 124.4 | 7.2 | 6.0 | 18.0 | 24.0 |
| | CTR | | 223.1 | 65.9 | 26.0 | 131.6 | 87.3 |

Table 3: Network size (#parameters in millions, regardless of trainable or non-trainable) and average training time per task of each model measured in minutes. We use "*" to indicate a replay method with a memory buffer. Here we report #parameters without including the memory buffer. The extra parameters and more training time used by our system are mainly due to the use of capsules and adapters.

training epochs. All reported test results in the paper are given by the parameters with the best validation performance.

# References

[1] J. Serrà, D. Suris, M. Miron, and A. Karatzoglou. Overcoming catastrophic forgetting with hard attention to the task. In *ICML*, 2018.