# OpenReview forum: "Achieving Forgetting Prevention and Knowledge Transfer in Continual Learning"
_NeurIPS.cc/2021/Conference — NeurIPS 2021 Poster_

### Official Review · Reviewer_Ckwk · 2021-06-25

**Rating:** 6
**Confidence:** 2

**Summary:**

This work studies the continual learning of a sequence of NLP tasks. A novel model called AFK is proposed, which contains a series of hand-crafted modules to prevent forgetting and help with knowledge transfer. A number of baselines are compared with, and strong CL performance are shown in the experiments.

**Limitations And Societal Impact:**

The authors discuss how the architecture could slow down training briefly in the conclusion. It would be helpful to give more details.

**Main Review:**

I'm mainly voting to accept this paper because the model architecture is novel, and experiment results are strong. However, I admit I'm not an expert in the CL field, so my judgement is not confident.

My major problem with the paper is the writing: especially section 4. The architecture seems overly complicated to me, I don't even totally understand them after I read them twice. I hope the authors can add more intuition or high-level descriptions for the model (like what's the reason behind this design?). It would also be good to show some examples about how the modules work in practice. (Figure 2B is a good example)

Questions:

Re "Each capsule in the TK-Layer represents a task, and it prepares the low-level features derived from each task", does that mean the capsule for the first will be frozen when training for other tasks?

Is the introduction of capsules necessary in the model design ? It seems to me most of the modules are standard NN modules, and the system is trained end-to-end. Could you give more intuition about how capsules work, and how they are different from other modules ? (I'm not very familiar with capsules.) For example, can I just understand the capsule as a MLP? If that's the case why you call it capsule ?

About presentation:
I think it would be helpful to provide something like average accuracy in Table to help readers. Also, it would be helpful to show how other methods "forgets" about previous tasks, while AFK can remember them in the trajectory of learning.

Citations:
This paper talks about knowledge transfer https://arxiv.org/abs/1910.07117 , from a digalogue model perspective.


**Time Spent Reviewing:**

2.5

---

> ### Author Response · Authors · 2021-08-09
> **Thank you for your valuable comments. Our responses are as follows**
>
> > 1. I hope the authors can add more intuition or high-level descriptions for the model (like what's the reason behind this design?). It would also be good to show some examples about how the modules work in practice. (Figure 2B is a good example)...Is the introduction of capsules necessary in the model design ? It seems to me most of the modules are standard NN modules, and the system is trained end-to-end. Could you give more intuition about how capsules work, and how they are different from other modules ? (I'm not very familiar with capsules.) For example, can I just understand the capsule as a MLP? If that's the case why you call it capsule ?”
>
> The architecture of the capsule network framework is quite suited for continual learning because it can be conveniently adapted to help prevent catastrophic forgetting and encourage knowledge transfer in continual learning. As mentioned in lines 96-107, capsule networks are different from conventional neural networks in two major aspects. First, there is additional information encoded in each capsule. In contrast to classic neural networks (for example, MLP as you mentioned), a capsule network replaces a neuron (which outputs a scalar) with a capsule (which outputs a vector). This is important because more information can be encoded in a vector (the vector has both length and orientation), which enables more reliable similarity comparison in our routing algorithm. For example, the orientation can encode the features of a task (e.g., the sentiment words of a task), while the length of the vector can reflect the probability of existence of the features (how likely a task has certain features).
>
> Second, as mentioned in lines 96-107, the capsule network with our new task routing algorithm can block dissimilar task capsules (by setting their gates to 0) and open similar task capsules (by setting their gates to 1) during training, which help avoid catastrophic forgetting while encouraging positive knowledge transfer. Specifically, the routing mechanism dynamically decides which task capsules to use (or to block) based on the similarity of features from task capsules per data instance, and then aggregates the features of unblocked task capsules to obtain a good task-shared representation.
>
> we will add some examples about how the modules work in the revised version of the paper.
>
> > 2. Each capsule in the TK-Layer represents a task, and it prepares the low-level features derived from each task", does that mean the capsule for the first will be frozen when training for other tasks?
>
> It is not frozen because the task routing mechanism can automatically block dissimilar task capsules (by setting their gates to 0) and open similar task capsules (by setting their gates to 1) during training to help avoid catastrophic forgetting and encourage knowledge transfer.
>
> > 3. Also, it would be helpful to show how other methods "forgets" about previous tasks, while AFK can remember them in the trajectory of learning.
>
> We can see forgetting by comparing the baseline models and the SDL (standalone learning of each task independently) results in Table 2. For instance, we can see the BERT NFH and Adapter-BERT NFH are much worse than SDL, indicating they suffer greatly from catastrophic forgetting. By comparing AFK and SDL, we can clearly see that AFK is doing better than SDL, indicating forgetting prevention and knowledge transferring.
>
> To make this clearer, we can add a figure of progressive results for AFK and baselines. For example, the table gives the first task’s macro F1 (MF1) score after each subsequent task is learned in the ASC experiment for AFK and Adapter-BERT DER++. DER++ is one of the latest systems published in NeurIPS-2020.
>
> | After task |   0  |   1  |   2  |   3  |   4  |   5  |   6  |   7  |   8  |   9  |  10  |  11  |  12  |  13  |  14  |  15  |  16  |  17  |  18  |
> |:----------:|:----:|:----:|:----:|:----:|:----:|:----:|:----:|:----:|:----:|:----:|:----:|:----:|:----:|:----:|:----:|:----:|:----:|:----:|:----:|
> |     AFK    | 78.9 | 88.9 | 80.2 | 80.2 | 87.9 | 78.9 | 83.1 | 83.1 | 83.1 | 93.4 | 93.4 | 93.4 | 93.4 | 87.9 | 87.9 | 87.9 | 93.4 | 93.4 | 87.9 |
> |    DER++   | 78.9 | 70.6 | 69.1 | 71.5 | 62.2 | 32.9 | 33.4 | 30.0 | 48.0 | 48.8 | 65.5 | 32.7 | 13.8 | 32.8 | 13.2 | 35.6 | 23.9 | 32.4 | 49.9 |
>
> For our AFK, we see that the performance of the first task is doing better and better over the learning of subsequent tasks, despite some small fluctuations in the middle. This indicates the effectiveness of (backward) knowledge transfer (for forward transfer, we can compare the results of AFK with the results of non-continual learning in Table 2). In contrast, DER++ is doing worse and worse over the learning of the subsequent, which shows that DER++ has severe forgetting.
>
> > 3. I think it would be helpful to provide something like average accuracy in Table to help readers. ”
>
> Thanks for the suggestion. We will provide it (adding another column in Table 2) in the revised version
>
> > 4. Citations: This paper talks about knowledge transfer https://arxiv.org/abs/1910.07117 , from a dialogue model perspective.”
>
> The proposed AFK focuses on preventing forgetting and encouraging transfer under continual learning using the capsule network and adapters. The mentioned paper is not about continual learning of a sequence of tasks. It tries to fine-tune a pre-trained model while also preventing forgetting of the pretrained knowledge. We will cite and discuss it in the revised paper.

---

> > ### Comment · Reviewer_Ckwk · 2021-08-22
> > **thanks**
> >
> > The response is helpful. THanks!

---

### Official Review · Reviewer_jqf7 · 2021-07-13

**Rating:** 6
**Confidence:** 4

**Summary:**

This paper studies the problem of continual learning in a popular NLP task, text classification.  It proposes a framework called AFK, that utilizes capsule networks to dynamically route task-specific knowledge among task-shared knowledge. The author claims two major benefits of doing this: (1) this essentially establishes an attention effect that allows the model to more effectively conduct forward transfer, and (2) the assignment of new parameters for new tasks and the masking mechanism can mitigate catastrophic forgetting. Overall, the application of capsule network in continual learning is novel and the framework obtains good performance. However, I feel that more comprehensive analysis could further supported the presentation, and some could be critical for understanding these models (see below).

**Limitations And Societal Impact:**

None.

**Main Review:**

Pros:
1. The proposed idea of using capsule network for better knowledge transfer in continual learning is novel.
2. Experimental results do show the efficacy of the method.
3. The paper includes an amazing amount of baselines, which could allow us to compare among them as well.

Cons:
1. Not too much analysis is included for understanding model behaviors. For example, since the method is designed to capture task transferability for better knowledge transfer, can you design some analysis to show this? Do similar tasks enjoy similar metrics? Can we tell which task(s) benefit a specific task?
2. Experimental settings can be more comprehensive. While a lot of baselines are included, interestingly a naive MTL model is not, which is usually considered as the upper bound for continual learning. In addition, the paper only conduct experiments on text classifications, and thus it is also interesting to explore other types or even mixture of different types of tasks, which should be more challenging and practical.
3. Motivation is not enough. I am not sure if I missed this, but why capsule network in particular for your AFK framework? Based on my understanding, you can also do, for example, a mixture-of-expert structure with adapters? It is of course nothing wrong to explore something new, but could you please give a bit more motivation for this? It would also be great if you can present some comparison with MOE too.

Questions:
1. Memory-replay-based methods are known to be robust to catastrophic forgetting. Do you have any idea why they didn't do well in your settings?
2. [1] also explores comparing forward and backward knowledge transfer. In particular, they found negative transfer to the latest task seen. Do you have similar observations?

Reference:
[1] Efficient Meta Lifelong-Learning with Limited Memory. Wang et al., EMNLP 2020

Some small suggestions:
1. Please highlight some results in Table 2. It is quite painful to search for numbers in a giant table like this.
2. I would recommend using a different name. AKF ((Achieving Forgetting avoidance and Knowledge transfer in continual learning) is too broad and doesn't sound like a method name to me.





**Time Spent Reviewing:**

2

---

> ### Author Response · Authors · 2021-08-09
> **Thank you for your valuable comments. Our responses are as follows**
>
> > 1. Not too much analysis is included for understanding model behaviors. For example, since the method is designed to capture task transferability for better knowledge transfer, can you design some analysis to show this? Do similar tasks enjoy similar metrics? Can we tell which task(s) benefit a specific task?
>
> Since our routing algorithm automatically detects similarity and transferability at the data instance level in the latent space, it is hard to know exactly which previous tasks have transferred knowledge to the new task. We humans probably behave similarly as we don’t always consciously know what we learned in the past has helped us learn a new task quickly. But this is an interesting topic to be studied in the future.
>
> For ASC and DSC specifically, they are two types of sentiment classification problems. Intuitively, we know that their tasks/domains have similarities because positive and negative sentiments are usually expressed with sentiment words/phrases, such as good and wonderful (positive), and bad and terrible (negative), and different tasks/domains share similar words/phrases to express the positive or negative sentiment.
>
> The following shows an example of knowledge transfer. The table gives the first task’s macro F1 (MF1) score after each subsequent task is learned in the ASC experiment.
>
> | After task |   0  |   1  |   2  |   3  |   4  |   5  |   6  |   7  |   8  |   9  |  10  |  11  |  12  |  13  |  14  |  15  |  16  |  17  |  18  |
> |:----------:|:----:|:----:|:----:|:----:|:----:|:----:|:----:|:----:|:----:|:----:|:----:|:----:|:----:|:----:|:----:|:----:|:----:|:----:|:----:|
> |     AFK    | 78.9 | 88.9 | 80.2 | 80.2 | 87.9 | 78.9 | 83.1 | 83.1 | 83.1 | 93.4 | 93.4 | 93.4 | 93.4 | 87.9 | 87.9 | 87.9 | 93.4 | 93.4 | 87.9 |
>
> We can see that the performance of the first task is doing better and better over the learning of subsequent tasks, despite some small fluctuations in the middle. This indicates the effectiveness of (backward) knowledge transfer. For forward transfer, we can compare the results of AFK with the results of non-continual learning in Table 2, which is discussed in line 311-318.
>
> > 2. Experimental settings can be more comprehensive. While a lot of baselines are included, interestingly a naive MTL model is not, which is usually considered as the upper bound for continual learning..
>
> We conducted the experiments on Multi-task Learning (MTL, specifically BERT MTL):
>
> | Dataset |  ASC  |  ASC  | DSC(200) | DSC(200) | DSC(full) | DSC(full) | 20News | 20News |
> |:-------:|:-----:|:-----:|:--------:|:--------:|:---------:|:---------:|:------:|:------:|
> |  Model  |  Acc. |  MF1  |   Acc.   |    MF1   |    Acc.   |    MF1    |  Acc.  |   MF1  |
> |   MTL   | 91.91 | 88.11 |   85.05  |   84.03  |   89.77   |   89.28   |  96.77 |  96.77 |
> |   AFK   | 89.47 | 83.62 |   84.34  |   83.29  |   89.31   |   88.75   |  95.25 |  95.23 |
>
>
> We can see that as the upper bound, MTL is slightly better than AFK, which also indicates AFK is highly effective in avoiding forgetting and encouraging transfer.
>
> > 3. In addition, the paper only conduct experiments on text classifications, and thus it is also interesting to explore other types or even mixture of different types of tasks, which should be more challenging and practical
>
> In the paper, we show that AFK is effective in learning similar tasks (ASC and DSC) and dissimilar tasks (20 Newsgroup). In our future work, we will investigate the learning of a mixture of different types of tasks. Our current work focuses on text classification as it uses language models. In the future, we will also try images and other forms of data.
>
> >4. Motivation is not enough. I am not sure if I missed this, but why capsule network in particular for your AFK framework? Based on my understanding, you can also do, for example, a mixture-of-expert structure with adapters? It is of course nothing wrong to explore something new, but could you please give a bit more motivation for this? It would also be great if you can present some comparison with MOE too.”
>
> As mentioned in lines 96-107, the capsule network with our new task routing algorithm is very useful in continual learning because the routing mechanism can block dissimilar task capsules (by setting their gates to 0) and open similar task capsules (by setting their gates to 1) during training, which helps avoid catastrophic forgetting while encouraging positive knowledge transfer. Specifically, the routing mechanism dynamically decides which task capsules to use (or to block) based on the similarity of features from the task capsules at the instance level, and then aggregates the features of unblocked task capsules to obtain a good task-shared representation. We use capsules as they contain more information than traditional neurons, which is important in our continual learning setting because they allow similarity comparison to be done more reliably.
>
> This may remind readers of the mixture of experts (MoE). Capsule networks and MoE are similar in that they both combine units (i.e., capsules or experts) specializing in different regions of the input space, and they adjust the contributions of units per sample. The key differences are: First, different capsules do not share the same feature space in capsule networks while different experts may have a shared feature space in MoE. This difference is important for continual learning because different feature spaces of capsule networks help avoid forgetting. Second, perhaps more importantly, the gating mechanism in the capsule networks can dynamically suppress (or open) dissimilar (or similar) low-level capsules via our routing algorithm to further help prevent forgetting and to enable knowledge transfer, but the current MoE cannot do that.
>
> > 5. Memory-replay-based methods are known to be robust to catastrophic forgetting. Do you have any idea why they didn't do well in your settings?”
>
> Replay-based methods have forgetting issues as well. For example, DER++ ([3] in the paper) also shows forgetting in their experiments. In general, replay-based methods have the risk of overfitting the small subset of the stored/replayed samples as the replayed data is the only information for the previous tasks used in training the new task. We can observe from Table 2 that parameter isolation (using masks) based methods such as HAT and CAT do better than replay-based methods A-GEM and DER++, which use regularization to prevent forgetting. This is perhaps because the masks can almost guarantee no forgetting, but regularization does not have this guarantee and may perform very well on some datasets but not so well on others. Furthermore, these systems were originally designed for image classification. They may not be suitable for text classification using pre-trained language models in text classification.
>
> > 6. [1] also explores comparing forward and backward knowledge transfer. In particular, they found negative transfer to the latest task seen. Do you have similar observations?”
>
> After reading the paper, we noticed that the observation in [1] is made about learning dissimilar tasks. In our case, this corresponds to the 20News data (which also have dissimilar tasks). Yes, we have the same observation. If we compare the result of AFK (forward) (which is the test performance when each task was first learned, the same setting as that in [1]) with the standalone training of each task (SDL) in Table 2, we can observe that the result of AFK (forward) is slightly poorer than SDL’s result for 20News. This is because dissimilar tasks have little shared knowledge among them, and for continual learning, the training of the new task must take a trade-off to balance the need for preventing forgetting of previous tasks and the need for achieving good results for the current new task. However, for similar tasks in the ASC and DSC experiments, we do not have the observation because their tasks are similar and have shared knowledge, and AFK also has the transfer routing mechanism to explicitly leverage the shared knowledge, which enables AFK (forward) to perform better than SDL.
>
>
> Thanks for the other suggestions. We will add highlights in Table 2 and also change the name of our system.

---

> > ### Comment · Reviewer_jqf7 · 2021-08-22
> > **Score updated**
> >
> > Thanks for your response. I think most of my concerns are addressed and thus I'd happy to update my score.

---

### Official Review · Reviewer_mCBm · 2021-07-15

**Rating:** 7
**Confidence:** 4

**Summary:**

The authors proposed a new continual learning method AFK, aiming to prevent catastrophic forgetting and exhibit knowledge transfer. Capsule Network is used to share task-specific knowledge in the continual learning process. Experimental results show that AFK outperforms previous SOTA approaches on many different continual learning sentiment classification datasets.

**Limitations And Societal Impact:**

The authors have adequately addressed the limitations (e.g. large capsule slow down training). I did not see any potential negative societal impact of their work.

**Main Review:**

In this paper, the authors aim to address not only catastrophic forgetting which is intensively explored in previous work but also knowledge transfer.
Using the routing algorithm in capsule network to encourage knowledge transfer is novel and reasonable. This may be the first paper to incorporate capsule network into continual learning (CL).
The results support the effectiveness of the proposed method which outperforms the other CL methods.
The writing of this paper is clear.

Suggestions:
1. In Table 2, it’s better to use bold numbers to highlight the best score in each part. A bunch of tiny digits in this large table is not easy for the reader to focus on the important parts.
2. In continual learning, we usually want to see the visualization of accuracy for each old task, which may go up and down in the whole training process. It allows us to observe the details of catastrophic forgetting/knowledge transfer for each method. If the authors can show the details with some figures, it would be easier for readers to grasp the experiment results.
3. For text classification, there is a strong continual learning method called LAMOL [1]. I suggest the authors also compared AFK with their results.
4. In Table 2, the authors only show non-CL (single-task) results and CL results. However, the multi-task learning model should also be considered as a strong upper bound of CL. It can tell us how well the model can achieve without the continual learning setting (under the same network capacity). I suggest that the authors can include the multi-task results using a model similar to the AFK model.
5. Capsule network and the routing algorithm are the backbone of the proposed method, but they are not shown in the title. I think it is better to use “capsule” in your title (or in the abbreviation AFK) to make it more clear and consistent with your proposed content.

[1] Fan-Keng Sun, Cheng-Hao Ho, Hung-Yi Lee. LAMOL: LAnguage MOdeling for Lifelong Language Learning. ICLR 2020.


**Time Spent Reviewing:**

3 hrs

---

> ### Author Response · Authors · 2021-08-09
> **Thank you for your valuable comments. Our responses are as follows**
>
> > 1. In continual learning, we usually want to see the visualization of accuracy for each old task, which may go up and down in the whole training process. It allows us to observe the details of catastrophic forgetting/knowledge transfer for each method. If the authors can show the details with some figures, it would be easier for readers to grasp the experiment results.
>
> Thanks for the suggestion. We will draw a figure in the revised version. As there are a large number of tasks, here we show a representative example. The following table gives the first task’s macro F1 (MF1) score after each subsequent task is learned in the ASC experiment.
>
> | After task |   0  |   1  |   2  |   3  |   4  |   5  |   6  |   7  |   8  |   9  |  10  |  11  |  12  |  13  |  14  |  15  |  16  |  17  |  18  |
> |:----------:|:----:|:----:|:----:|:----:|:----:|:----:|:----:|:----:|:----:|:----:|:----:|:----:|:----:|:----:|:----:|:----:|:----:|:----:|:----:|
> |     AFK    | 78.9 | 88.9 | 80.2 | 80.2 | 87.9 | 78.9 | 83.1 | 83.1 | 83.1 | 93.4 | 93.4 | 93.4 | 93.4 | 87.9 | 87.9 | 87.9 | 93.4 | 93.4 | 87.9 |
>
> We can see that the performance of the first task is doing better and better over the learning of subsequent tasks, despite some small fluctuations in the middle. This indicates the effectiveness of (backward) transfer.
>
> > 2. For text classification, there is a strong continual learning method called LAMOL [1]. I suggest the authors also compared AFK with their results .... However, the multi-task learning model should also be considered as a strong upper bound of CL. It can tell us how well the model can achieve without the continual learning setting (under the same network capacity). I suggest that the authors can include the multi-task results using a model similar to the AFK model.
>
> Thanks for the suggestion, we have conducted experiments on LAMOL and Multi-task Learning (MTL, specifically BERT MTL):
>
> | Dataset |  ASC  |  ASC  | DSC(200) | DSC(200) | DSC(full) | DSC(full) | 20News | 20News |
> |:-------:|:-----:|:-----:|:--------:|:--------:|:---------:|:---------:|:------:|:------:|
> |  Model  |  Acc. |  MF1  |   Acc.   |    MF1   |    Acc.   |    MF1    |  Acc.  |   MF1  |
> |   MTL   | 91.91 | 88.11 |   85.05  |   84.03  |   89.77   |   89.28   |  96.77 |  96.77 |
> |  LAMOL  | 88.91 | 80.59 |   89.12  |   86.58  |   92.11   |   91.72   |  66.13 |  45.74 |
> |   AFK   | 89.47 | 83.62 |   84.34  |   83.29  |   89.31   |   88.75   |  95.25 |  95.23 |
>
>
> Note that LAMOL is based on GPT-2, which is known to be a more powerful model than BERT (also noted and shown in the LAMOL paper [1]). This explains why LAMOL outperforms even BERT-based MTL in DSC. However,  AFK outperforms LAMOL in ASC and 20News even with the less powerful BERT model that AFK adopts. ASC is a more difficult and complex task because it classifies each sentence which contains much less information than a document in DSC. This indicates that AFK has the ability to deal with a difficult task in comparison with LAMOL. Regarding the 20News, as we mentioned in lines 234-239, its tasks are very different/dissimilar and thus have little knowledge to be transferred across tasks. Dealing with forgetting is the main issue. We can see that LAMOL is quite poor compared to AFK in this case. LAMOL is a pseudo-replay method, which may generate poor pseudo examples and result in forgetting. AFK has almost no forgetting.
>
> Regarding the MTL model (the upper bound), we can see it is slightly better than AFK, which again shows that AFK is highly effective in overcoming forgetting and encouraging knowledge transfer.
>
> Thanks for the other suggestions. We will add highlights in Table 2 and also change the name of our system.

---

> > ### Comment · Reviewer_mCBm · 2021-08-25
> > **Thanks. Good work!**
> >
> > Thanks for your response! Good to see these experiments that address my concerns. Well done!

---

### Official Review · Reviewer_fMez · 2021-07-21

**Rating:** 6
**Confidence:** 5

**Summary:**

The paper investigates the problem of sequential learning of NLP tasks. Concretely, it proposes a novel method, AFK, to alleviate catastrophic forgetting and enable knowledge transfer across related tasks. The paper argues that most of the existing works solely focus on catastrophic forgetting problems and there is no explicit mechanism to facilitate the transfer across tasks. To address this issue, the paper designs a novel architecture as a part of the AFK method, CBA (Continual learning BERT Adapter). CBA consists of a capsule network to model different tasks and a transfer routing algorithm to support knowledge transfer across tasks. For experimentation, the paper considers three text classification datasets - Document Sentiment Classification (10 tasks), Aspect Sentiment Classification (19 tasks), and 20News (10 tasks) and evaluates AFK on task-incremental learning scenarios (separate classification head per task). In comparison with the considered baselines, the paper reports improved overall performance and minimal forgetting for AFK.

**Ethical Concerns:**

No!

**Limitations And Societal Impact:**

Yes, the paper discusses the limitations of their work.


**Main Review:**

Naive fine-tuning of the BERT model is known to undergo forgetting. To address this issue, several works have been proposed with Adapters being more compact and modular. Building upon this line of work, the paper proposes a continual learning variant of the BERT Adapter (CBA). Instead of a simple feedforward bottleneck layer in classical Adapters, the paper introduces well-motivated layers to CBA: (i) a capsule per task (TK-layer), (ii) a transfer capsule layer (TR-layer) mainly consisting of similarity estimator and task router to control the influence of the previous tasks while learning the current task, (iii) task-specific modulation layer to mask neurons corresponding to the previous layer, thus, minimizing the forgetting. However, based upon the similar motivation to AFK, i.e, using knowledge from the previous tasks while learning the current task, AdapterFusion [1] work proposes to learn a task-specific composition of adapters from previous tasks. The paper completely ignores this relevant line of work and ends up just comparing with a single adapter per task or applying existing CL methods to vanilla adapters.

In section 5.4, the paper compares standalone training (SDL) with NFH (continual learning with no forgetting handled). Based upon this comparison, the paper suggests that ASC and DSC tasks have similarities. However, there is no explicit metric to quantify this similarity (for example, some form of clustering on representations extracted from pre-trained BERT models?). The paper seems to make an implicit assumption that similar tasks undergo less forgetting. Based on the recent controlled study [4], the relationship between task similarity and forgetting is less straightforward. The paper should review this work and reformulate its claims.

The paper uses the term Forward Transfer without any formal definition. Based upon the textual description it seems that the paper wanted to refer to the Learning Accuracy metric which is considered in the literature. Moreover, in the continual learning literature, Forward Transfer refers to the zero-shot evaluation of the model from previous tasks. Please review [2, 3] to be consistent with the existing terminology and restrain from overloading existing terms to avoid confusion.

The paper motivates that there is a need to explicitly model knowledge transfer between tasks. However, it is unclear from Table 2 whether there is any positive backward transfer happening. At least AdapterFusion guarantees that there would be no forgetting (refer [2] for the definition of the backward transfer). Therefore, it is important to lay down a clear distinction with the existing work and explicitly focus on analysis to validate the proposition being in the paper.

Overall, the paper attempts to solve an important problem of continual learning in the context of the pre-trained Transformers. With the rise of larger models, it is important to consider the training and inference time of the methods. Given the paper builds upon the parameter-efficient transfer learning, the number of parameters would not grow dramatically with the number of tasks. However, with the use of a capsule network, the proposed approach is expensive and the paper rightly acknowledges it to be the limitation of their work.

[1] Pfeiffer, Jonas, et al. "AdapterFusion: Non-Destructive Task Composition for Transfer Learning." Proceedings of the 16th Conference of the European Chapter of the Association for Computational Linguistics: Main Volume. 2021.

[2] Riemer, Matthew, et al. "Learning to Learn without Forgetting by Maximizing Transfer and Minimizing Interference." International Conference on Learning Representations. 2018.

[3] Lopez-Paz, David, and Marc'Aurelio Ranzato. "Gradient episodic memory for continual learning." Advances in neural information processing systems 30 (2017): 6467-6476.

[4] Ramasesh, Vinay Venkatesh, Ethan Dyer, and Maithra Raghu. "Anatomy of Catastrophic Forgetting: Hidden Representations and Task Semantics." International Conference on Learning Representations. 2020.

Typos: In Eq (6), should it be $u_{j|t}^{(t)}$ instead of $v_{j|t}^{(t)}$.



**Time Spent Reviewing:**

4

---

> ### Author Response · Authors · 2021-08-09
> **Thank you for your valuable comments. Our responses are as follows**
>
> > 1. AdapterFusion [1] work proposes to learn a task-specific composition of adapters from previous tasks. The paper completely ignores this relevant line of work and ends up just comparing with a single adapter per task or applying existing CL methods to vanilla adapters.
>
> Sorry, we missed the AdapterFusion work, which is relevant as it also uses adapters like our AFK. However, AdapterFusion is not about continual learning. Specifically, AdapterFusion proposes a two-stage method to learn a set of tasks. In the first stage, it learns one adapter for each task independently using the task’s training data. In the second stage, it uses the training data again to learn a good composition of the learned adapters in the first stage to produce the final model for all tasks. AdapterFusion basically tries to improve multi-task learning. However, AFK is a continual learning system that learns a sequence of tasks incrementally one by one, and it prevents forgetting and encourages knowledge transfer using the capsule network and a novel transfer routing algorithm. Thus the two systems have different learning settings. We will cite and compare AdapterFusion in our revised paper.
>
> > 2. Based upon this comparison, the paper suggests that ASC and DSC tasks have similarities. However, there is no explicit metric to quantify this similarity (for example, some form of clustering on representations extracted from pre-trained BERT models?). The paper seems to make an implicit assumption that similar tasks undergo less forgetting. Based on the recent controlled study [4], the relationship between task similarity and forgetting is less straightforward. The paper should review this work and reformulate its claims.
>
> ASC and DSC are two types of sentiment classification problems. Their tasks/domains have similarities because positive or negative sentiments are usually expressed with sentiment words/phrases, such as good and wonderful (positive), and bad and terrible (negative), and different tasks/domains use similar words/phrases to express positive or negative sentiment.
>
> About the controlled study in [4], the authors observed that similar tasks lead to less forgetting in the sequential binary classification tasks setting (6.1 Setup 1). The only opposite observation is under a very different setting, i.e., the first task has 4 classes but the second task has only two classes (6.1 Setup 2). This inconsistent number of classes may result in different optimization processes and destroy the similar representations for similar tasks. AFK works in a similar setting to Setup 1 in [4].
>
> Explicitly measuring task similarity is an interesting topic. We will investigate this issue and your suggestions in our future work. The controlled study paper [4] also did not give an explicit metric, probably due to its difficulty.
>
> > 3. The paper uses the term Forward Transfer without any formal definition. Based upon the textual description it seems that the paper wanted to refer to the Learning Accuracy metric which is considered in the literature. Moreover, in the continual learning literature, Forward Transfer refers to the zero-shot evaluation of the model from previous tasks. Please review [2, 3] to be consistent with the existing terminology and restrain from overloading existing terms to avoid confusion.
>
> In our work, forward performance is the performance of a task when it was first learned, measuring how well a new task can make use of its previous knowledge to learn better. Forward transfer is defined as the forward performance subtracting the standalone (SDL) result. Note that this is not the only way to measure forward transfer. In references [2, 3] that you mentioned, forward transfer is measured by comparing the test results of the new task on the current learned network and a random initialized network before training the new task (thus it is zero-shot). This method indicates whether the learned network contains some useful knowledge for the new task. However, it does not tell how much forward transfer actually happens after learning the new task, which is what our method does. We believe that our measure is more useful in practice (which is also used in CAT ([22] in the paper). So although their metric and our metric both use the term forward transfer, the two metrics measure different things. We will include these and cite [2, 3] in the revised paper.
>
> > 4. It is unclear from Table 2 whether there is any positive backward transfer happening. At least AdapterFusion guarantees that there would be no forgetting (refer [2] for the definition of the backward transfer).
>
> In our work, the backward performance is the final performance after all tasks are learned. Backward transfer is defined as the difference between the backward performance and the forward performance. This measure is also used in [2, 3] (Eq. 3 in [3] and [2] uses the same Eq as [3]). If we compare the final (or backward) performance of AFK (last row) and that of AFK (forward) in Table 2, we can see whether backward transfer happens on average of all tasks. We discussed the backward transfer performance of Table 2 in lines 310-318.
>
> We can also see the effect of backward transfer on a single task. For example, the table below shows how the macro F1 (MF1) of the first task of ASC changes after each subsequent task is trained.
>
> | After task |   0  |   1  |   2  |   3  |   4  |   5  |   6  |   7  |   8  |   9  |  10  |  11  |  12  |  13  |  14  |  15  |  16  |  17  |  18  |
> |:----------:|:----:|:----:|:----:|:----:|:----:|:----:|:----:|:----:|:----:|:----:|:----:|:----:|:----:|:----:|:----:|:----:|:----:|:----:|:----:|
> |     AFK    | 78.9 | 88.9 | 80.2 | 80.2 | 87.9 | 78.9 | 83.1 | 83.1 | 83.1 | 93.4 | 93.4 | 93.4 | 93.4 | 87.9 | 87.9 | 87.9 | 93.4 | 93.4 | 87.9 |
>
>
> We can see from the above Table that the performance of the first task is in general doing better and better after more tasks are learned despite some fluctuations in the middle. This indicates the effectiveness of backward transfer.
>
> As we discussed above, AdapterFusion is not a continual learning method and is more like a multi-task learning method but using two steps, and it thus has no forgetting. Unlike AdapterFusion, our AFK is a continual learning method that learns a sequence of tasks incrementally and after each task is learned its trained data is no longer accessible.
>
>
> Thanks for the other suggestions. We will fix all the typos in the revised version.

---

> > ### Comment · Reviewer_fMez · 2021-09-01
> > **Thanks for the response.**
> >
> > Thanks for clarifying your definition of the forward transfer and how it differs from the literature. Also, regarding the backward transfer, I expect authors to explicitly state them in the next version of the paper. Furthermore, comparisons with AdapterFusion should definitely improve this work.

---

### Decision · Program_Chairs · 2021-09-27

**Decision:**

Accept (Poster)

**Comment:**

The paper studies continually learning of a sequence of natural language processing (NLP) tasks. The authors claim that state of the art approaches that facilitate knowledge transfer (e.g. fine-tuning a BERT-like language model) suffer from serious catastrophic forgetting in the continual learning setting and present a novel model called AFK to achieve knowledge transfer while avoiding catastrophic forgetting in NLP tasks. AFK  makes use of a combination of Capsule networks to model each task, a transfer routing algorithm to identify and transfer knowledge across tasks to improve accuracy and task masks to prevent catastrophic forgetting. The effectiveness of the proposed method is demonstrated in empirical evaluations.

The reviewers raised some points that were satisfactorily addressed in the rebuttal, including additional experiments. The rebuttal helped convince the reviewers of the merits of the paper.